# OBS-Diff: Accurate Pruning For Diffusion Models in One-Shot

**Junhan Zhu[1], Hesong Wang[1,2], Mingluo Su[1], Zefang Wang[1,2], Huan Wang[1]***

[1]Westlake University    [2]Zhejiang University

https://github.com/Alrightlone/OBS-Diff

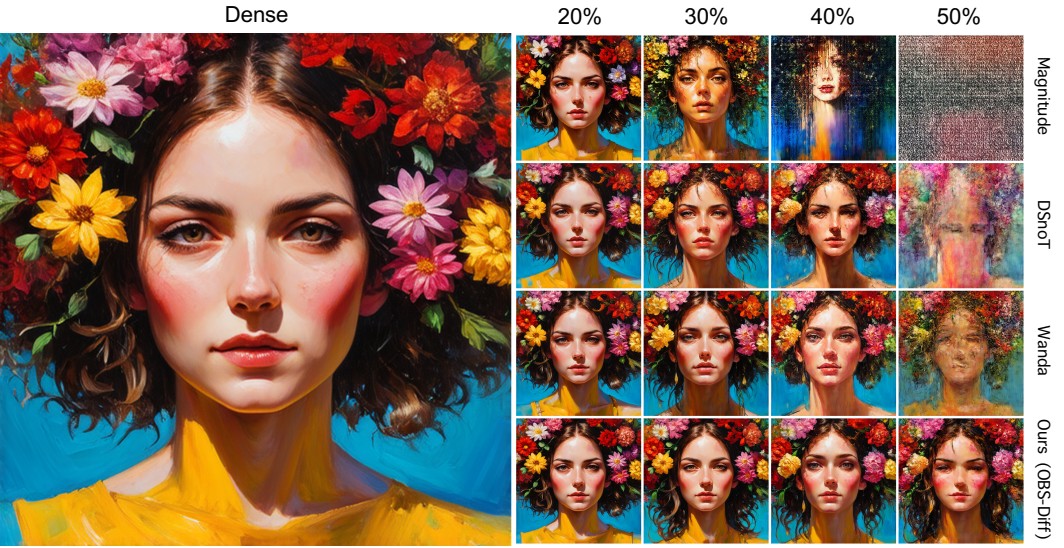

**Prompt**: A portrait of a human growing colorful flowers from her hair. Hyperrealistic oil painting. Intricate details.
**Negative Prompt**: low quality, bad quality, blur, ugly, worst quality

Figure 1: Qualitative comparison of unstructured pruning methods on the SD3-Medium model (Esser et al., 2024). We evaluate Magnitude, DSnoT (Zhang et al., 2024d), Wanda (Sun et al., 2024), and our method (OBS-Diff) at various sparsity levels (20%, 30%, 40%, and 50%) using the same prompt and negative prompt. All images are generated at a resolution of $512 \times 512$.

## Abstract

Large-scale text-to-image diffusion models, while powerful, suffer from prohibitive computational cost. Existing one-shot network pruning methods can hardly be directly applied to them due to the iterative denoising nature of diffusion models. To bridge the gap, this paper presents *OBS-Diff*, a novel one-shot pruning framework that enables accurate and training-free compression of large-scale text-to-image diffusion models. Specifically, **(i)** OBS-Diff revitalizes the classic Optimal Brain Surgeon (OBS), adapting it to the complex architectures of modern diffusion models and supporting diverse pruning granularity, including unstructured, N:M semi-structured, and structured (MHA heads and FFN neurons) sparsity; **(ii)** To align the pruning criteria with the iterative dynamics of the diffusion process, by examining the problem from an error-accumulation perspective, we propose a novel timestep-aware Hessian construction that incorporates a logarithmic-decrease weighting scheme, assigning greater importance to earlier timesteps to mitigate potential error accumulation; **(iii)** Furthermore, a computationally efficient group-wise sequential pruning strategy is proposed to amortize the expensive calibration process. Extensive experiments show that OBS-Diff achieves state-of-the-art one-shot pruning for diffusion models, delivering inference acceleration with minimal degradation in visual quality.

---

*Corresponding author: wanghuan@westlake.edu.cn

# 1 INTRODUCTION

Recent advances in text-to-image generation have been largely driven by large-scale diffusion models (Rombach et al., 2022; Ramesh et al., 2022; Xie et al., 2024). These models, such as the Stable Diffusion 3 and 3.5 series (Esser et al., 2024), are capable of producing stunning images from textual prompts, revolutionizing fields from digital art to content creation. However, their massive parameter counts—often in the billions (e.g., 8B in Stable Diffusion 3.5-Large)—create prohibitive computational and memory demands, severely limiting their broader accessibility.

To improve the efficiency of the diffusion models, multiple research avenues have been proposed. One major line of work focuses on accelerating the sampling process by reducing the number of denoising steps (Song et al., 2021; Lu et al., 2022; Wang & Wang, 2025) or knowledge distillation (Salimans & Ho, 2022; Sauer et al., 2024). Orthogonal to these efforts, model compression aims to reduce the intrinsic computational and memory footprint of the model itself. This category includes methods like quantization (He et al., 2023; Li et al., 2023b; Shang et al., 2023; Li et al., 2023a) and pruning, which is the primary focus of our work.

Pruning (Han et al., 2015; 2016; Wang et al., 2021; Fang et al., 2023a; Feng et al., 2024) is an effective way to reduce the computational and memory costs of deep neural networks. However, the rapid evolution of diffusion models underscores the severe limitations of existing pruning techniques. Current methods often lack generality (Fang et al., 2023b; Li et al., 2023c; Kim et al., 2024), as they are typically tailored to specific architectures like the U-Net and are not easily adapted to large-scale, text-to-image diffusion models with diverse structures (e.g., Multimodal Diffusion Transformer). Moreover, the efficiency gains from pruning are frequently undermined by computationally expensive requirements, such as the need for gradient information during pruning (Fang et al., 2023b; Zhang et al., 2024b) or a costly post-pruning fine-tuning stage. Furthermore, unstructured and semi-structured pruning remains largely unexplored for large-scale text-to-image diffusion models. All of these motivate our central research question: **Can we develop a general and training-free pruning framework capable of pruning diffusion models with diverse architectures and supporting multiple pruning granularities in a one-shot manner?**

In the domain of Large Language Models (LLMs), one-shot and training-free pruning methods like SparseGPT (Frantar & Alistarh, 2023) and Wanda (Sun et al., 2024) have achieved remarkable success. Subsequent SlimGPT (Ling et al., 2024) and SoBP (Wei et al., 2024) further explored training-free structured pruning. These layer-wise post-training pruning method approaches efficiently compress massive models without requiring costly retraining. However, the existing training-free pruning methods from the LLM field, such as SparseGPT, cannot be directly applied to the diffusion model. This is due to the unique challenges posed by diffusion models: their iterative nature, where parameters are shared across multiple denoising steps. Furthermore, the complex architectures introduce additional difficulties for pruning.

To bridge this gap, we introduce **OBS-Diff**, a novel one-shot, training-free pruning framework designed specifically for large-scale text-to-image diffusion models. Our approach revitalizes the classic Optimal Brain Surgeon (OBS) (Hassibi et al., 1992) and tailors it to the unique, iterative nature of the diffusion denoising process. By reformulating the pruning objective to account for the temporal dynamics of generation and introducing a computationally efficient calibration strategy, OBS-Diff efficiently removes redundant weights with minimal impact on performance, all without requiring any training during the pruning process or fine-tuning.

Our contributions are summarized as below:

- We adapt the OBS framework to handle the complex architectures of modern diffusion models, such as the Multimodal Diffusion Transformer (MMDiT), and demonstrate OBS-Diff versatility across unstructured, semi-structured (e.g., 2:4 sparsity patterns), and structured pruning (e.g., removing entire attention heads or FFN neurons).

- Recognizing that errors introduced in the early stages of the iterative denoising process have a compounding effect, we propose a **Timestep-Aware Hessian Construction**. This novel construction weights the importance of parameters according to their influence across the entire denoising trajectory, prioritizing the more sensitive early steps through a logarithmic weighting scheme.

- To overcome the prohibitive cost of sequential calibration in iterative models, we devise a **group-wise sequential pruning** strategy built upon "Module Packages". This approach amortizes the

expensive data collection process by processing layers in batches, striking an effective balance between computational time and memory requirements for the pruning process.

- Extensive experiments demonstrate that OBS-Diff sets a new state-of-the-art for training-free diffusion model pruning. It achieves inference acceleration while maintaining high visual quality, outperforming other layer-wise pruning methods across various sparsity levels and patterns.

## 2 RELATED WORK

**Pruning for Diffusion Models.** Several methods have explored pruning for diffusion models. An early approach, Diff-pruning (Fang et al., 2023b), introduced a gradient-based method for structured pruning. However, its demonstration on small-scale, non-text-to-image models (e.g., DDPMs) and its dependency on expensive retraining limit its applicability to modern, large-scale systems with diverse architectures. A significant line of research has since focused on compressing the UNet-based text-to-image diffusion models, with works like SnapFusion (Li et al., 2023c), MobileDiffusion (Zhao et al., 2024), BK-SDM (Kim et al., 2024), LAPTOP-Diff (Zhang et al., 2024a), and LD-Pruner (Castells et al., 2024) all targeting less salient components of the UNet architecture.

Other works have explored different architectures or techniques; for instance, Tinyfusion (Fang et al., 2025a) introduced depth pruning for the DiT architecture. More recently, EcoDiff (Zhang et al., 2024b) introduced a general pruning framework for text-to-image models applicable to diverse architectures; however, it remains dependent on a costly training phase to learn a pruning mask and requires extensive hyperparameter tuning. A common theme among these methods is a dependency on training or fine-tuning and a primary focus on architecture-specific, structured pruning. Furthermore, unstructured and semi-structured pruning for large-scale, text-to-image diffusion models remains a largely unexplored area.

**Layer-Wise Pruning Methods.** Early post-training compression methods, notably Optimal Brain Damage (OBD) (LeCun et al., 1989) and Optimal Brain Surgeon (OBS) (Hassibi et al., 1992), utilized Hessian-based saliency scores to prune individual weights. However, the prohibitive cost of computing and storing the full Hessian matrix limited their scalability. This challenge spurred the development of layer-wise approaches such as L-OBS (Dong et al., 2017) and Optimal Brain Compression (OBC) (Frantar & Alistarh, 2022), which approximate the Hessian locally to make pruning tractable.

As models scaled to billions of parameters, particularly in Large Language Models (LLMs), new methods emerged. SparseGPT (Frantar & Alistarh, 2023), Wanda (Sun et al., 2024), and DSnoT (Zhang et al., 2024d) focused on efficient unstructured and semi-structured (N:M pattern) pruning. Subsequently, SlimGPT (Ling et al., 2024) and SoBP (Wei et al., 2024) extended the OBS methodology to a structured granularity. Nevertheless, this family of compression methods remains unexplored in the field of diffusion models.

## 3 PRELIMINARIES

### 3.1 LAYER-WISE POST-TRAINING PRUNING

Post-training pruning often decomposes the global network compression problem into a series of independent, layer-wise subproblems (Hubara et al., 2021; Nagel et al., 2020; Aghasi et al., 2017). For each layer $l$, the objective is to find a pruned weight matrix $\hat{\mathbf{W}}_l$ that minimizes the output reconstruction error, given input activations $\mathbf{X}_l$ and a target sparsity $S_l$. This is formulated as:

$$\text{argmin}_{\hat{\mathbf{W}}_l} \left\| \mathbf{W}_l \mathbf{X}_l - \hat{\mathbf{W}}_l \mathbf{X}_l \right\|_2^2 \quad \text{s.t.} \quad \text{sparsity}(\hat{\mathbf{W}}_l) = S_l, \tag{1}$$

where $\| \cdot \|_2^2$ is the squared Euclidean norm. The network is pruned by sequentially solving this optimization problem for each layer.

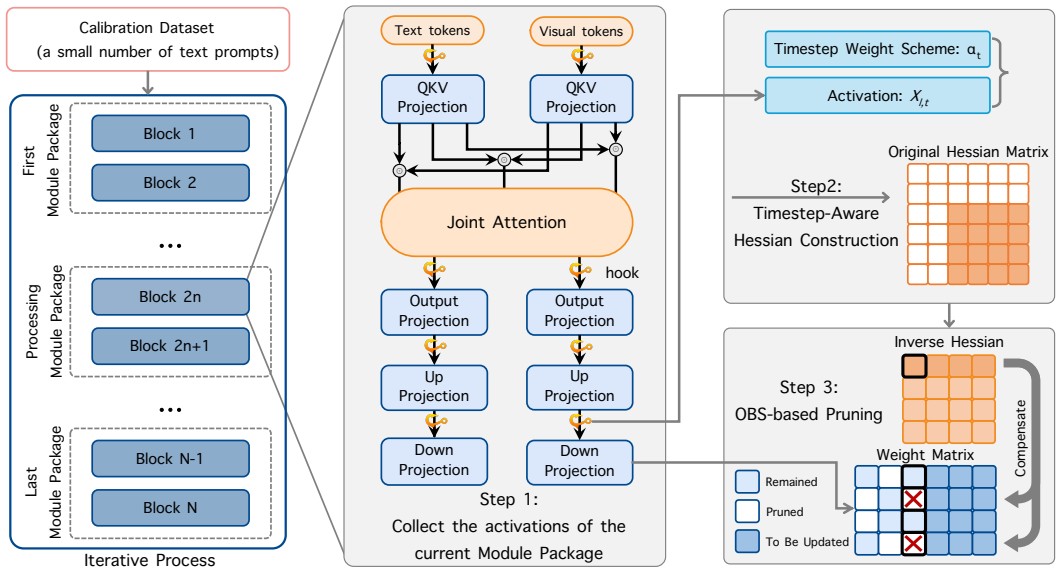

Figure 2: Illustration of the proposed **OBS-Diff** framework applied to the MMDiT architecture. Target modules are first partitioned into a predefined number of "Module Packages" and processed sequentially. For each package, hooks capture layer activations during a forward pass with a calibration dataset. This data, combined with weights from a dedicated timestep weighting scheme, is used to construct Hessian matrices. These matrices guide the Optimal Brain Surgeon (OBS) algorithm to simultaneously prune all layers within the current package before proceeding to the next.

## 3.2 OPTIMAL BRAIN SURGEON FOR LAYER-WISE PRUNING

The Optimal Brain Surgeon (OBS) framework (Hassibi et al., 1992) offers an efficient solution to the layer-wise problem in Eq. (1). A key insight of OBS is that the $\ell_2$-norm objective allows the problem to be decoupled into independent subproblems for each row of the weight matrix $\mathbf{W}_l$.

For each row, OBS approximates the objective with a second-order Taylor expansion centered around the current weights. This relies on the Hessian of the reconstruction error, $\mathbf{H} = 2\mathbf{X}_l\mathbf{X}_l^T$. This approximation yields a closed-form solution to identify the least salient weight $w_q$—the one whose removal minimally increases the error—and to compute the optimal update $\delta\mathbf{w}$ for the remaining weights in its row. The saliency score $\mathcal{L}_q$ and the update are defined as:

$$\mathcal{L}_q = \frac{w_q^2}{2[\mathbf{H}^{-1}]_{qq}}, \quad \delta\mathbf{w} = -\frac{w_q}{[\mathbf{H}^{-1}]_{qq}}\mathbf{H}_{:,q}^{-1}, \tag{2}$$

where $[\mathbf{H}^{-1}]_{qq}$ is the $q$-th diagonal element of the inverse Hessian and $\mathbf{H}_{:,q}^{-1}$ is its $q$-th column.

This process is repeated iteratively until the target sparsity $S_l$ is reached. After each weight is removed, the inverse Hessian must be updated. To circumvent the prohibitive cost of full re-inversion and the error accumulation of approximate rank-one updates, SparseGPT (Frantar & Alistarh, 2023) imposes a fixed pruning order. This structural constraint enables efficient and stable updates to the inverse Hessian information using methods like Cholesky decomposition (Frantar et al., 2024) as weights are progressively removed.

## 4 METHODOLOGY

We propose **OBS-Diff**, a one-shot, training-free pruning framework tailored for diffusion models. As illustrated in Figure 2, our method partitions the model into sequential "Module Packages" to amortize calibration costs. Within each package, we employ a novel Timestep-Aware Hessian construction to prioritize early denoising steps, enabling the simultaneous pruning of all target layers within the current package.

### 4.1 Timestep-Aware Hessian Construction

The layer-wise pruning objective defined in Eq. (1) is effective for models with a single forward pass, but insufficient for diffusion models, which are iterative and operate over a denoising trajectory parameterized by discrete timesteps $t \in \{1, \cdots, T\}$.[1] The impact of pruning-induced errors is not uniform across this trajectory. Errors introduced in early inference steps (small t) are inherently more damaging, as they propagate and compound through all subsequent steps $(t + 1, \cdots, T)$, leading to larger deviations in the final output.

Therefore, a robust pruning strategy must prioritize preserving network function during these critical early stages. We reformulate the layer-wise optimization problem to minimize a weighted reconstruction error that places greater importance on earlier, higher-impact steps:

$$\arg\min_{\hat{W}_l} \mathbb{E}_{t \sim [1,T]} \left[ \alpha_t \left\| W_l X_{l,t} - \hat{W}_l X_{l,t} \right\|_2^2 \right], \tag{3}$$

Here, $X_{l,t}$ is the input to layer $l$ at step $t$, and $\alpha_t$ is a step-dependent weight. We define $\alpha_t$ using a simple and effective logarithmically decreasing schedule:

$$\alpha_t = \alpha_{\min} + \frac{\alpha_{\max} - \alpha_{\min}}{\ln(T)} \ln(T - t + 1), \quad t \in \{1, 2, \cdots, T\}. \tag{4}$$

This schedule ensures the weight is highest at the beginning of inference and decays smoothly, such that $\alpha_1 > \alpha_2 > \cdots > \alpha_T > 0$.

By incorporating this weighting, we adapt the Optimal Brain Surgeon framework (Hassibi et al., 1992). The Hessian, which captures the second-order information of this weighted loss, is now computed as a weighted sum over all inference steps:

$$H_l = 2 \sum_{t=1}^{T} \alpha_t \mathbb{E}[X_{l,t} X_{l,t}^T], \tag{5}$$

which is termed as *Timestep-Aware Hessian*. It encapsulates the varying importance of parameters over the generation process. Saliency scores derived from its inverse are thus more sensitive to weights that are critical during the early, formative stages of the denoising process, resulting in a more faithfully pruned model.

### 4.2 Module Packages: A Group-wise Sequential Pruning Strategy

Conventional post-training pruning methods, such as SparseGPT (Frantar & Alistarh, 2023), employ a sequential layer-wise calibration. This paradigm is computationally prohibitive for diffusion models, as calibrating each layer necessitates executing a full, multi-step denoising trajectory. To address this bottleneck, we introduce **Module Packages**, a group-wise strategy that amortizes calibration costs by processing layers in batches.

Our approach is built upon two concepts. A **Basic Unit** is a set of layers with mutually independent inputs in a forward pass (e.g., query, key, and value projections), allowing for parallel processing. A **Module Package** comprises one or more Basic Units, which are pruned and calibrated collectively. Our framework processes these packages sequentially. For each package, we first execute a *Group-wise Data Collection* phase: we run the complete denoising trajectory once across the calibration dataset, using forward hooks to concurrently gather input statistics for all modules within the package. Subsequently, all modules are pruned simultaneously using their respective Timestep-Aware Hessian matrices.

Crucially, the network state is updated sequentially *between* packages but remains static *within* a package during data collection. This preserves the principle of sequential calibration at a coarser, group-wise granularity, rendering the process computationally feasible. This strategy drastically reduces the number of calibration runs, with the primary trade-off being an increased memory footprint to store multiple Hessian matrices concurrently. Notably, our empirical results demonstrate that pruning accuracy has low sensitivity to package granularity, granting practitioners the flexibility to balance computational cost against memory constraints without a significant performance sacrifice.

---

[1]Here, $t$ denotes the sequential index of the denoising iteration during inference, where $T$ is the total number of inference steps (e.g., for $T = 28$, $t$ ranges from $1, 2, \cdots, 28$).

### 4.3 EXTENSION TO SEMI-STRUCTURED AND STRUCTURED PRUNING

A key advantage of our OBS-Diff framework is its adaptability. While focusing on unstructured pruning, it readily extends to both semi-structured and structured sparsity.

**Semi-Structured Pruning.** For semi-structured patterns like 2:4 sparsity, the extension is direct. Within each block of four weights, we simply prune the two with the lowest per-weight OBS-Diff saliency scores, efficiently creating hardware-friendly models.

**Structured Pruning.** For structured pruning of Feed-Forward Network (FFN) layers, we assess a neuron's importance by aggregating the saliency of its associated weights. The saliency $\mathcal{L}_q$ for an entire neuron (column $q$) and the corresponding weight update are:

$$\mathcal{L}_q = \frac{\sum W_{:,q}^2}{2[\mathbf{H}^{-1}]_{qq}}, \quad \delta\mathbf{W} = -\frac{W_{:,q}}{[\mathbf{H}^{-1}]_{qq}}\mathbf{H}_{:,q}^{-1}, \tag{6}$$

where the lowest-scoring neurons are removed.

Similarly, for Multi-Head Attention (MHA), we prune entire heads. Our approach, inspired by SlimGPT (Ling et al., 2024), quantifies the saliency of each head.

The calculation begins with the full Hessian matrix, $\mathbf{H}$, for the output projection layer. For the $j$-th head, we consider its weight matrix $\mathbf{W}_j$ and the corresponding Hessian block $\mathbf{H_j}$. The total saliency for this head, $\mathcal{L}_j$, is found by aggregating the importance of its individual weights. The saliency is calculated as:

$$\mathcal{L}_j = \sum_{k=1}^{d} \frac{\sum (\mathbf{W}_j)_{:,k}^2}{(\mathbf{H_j}^{-1})_{k,k}}, \tag{7}$$

where $(\mathbf{W}_j)_{:,k}$ is the $k$-th column of the weight matrix $\mathbf{W}_j$, $(\mathbf{H_j}^{-1})_{k,k}$ is the $k$-th diagonal element of the inverse Hessian block, and $d$ is the dimension of each head.

However, MMDiT's joint attention mechanism presents a unique challenge. Shared attention heads process concatenated multi-modal inputs, but are fed into separate, modality-specific output paths. This structure yields two distinct importance rankings for the same set of heads (one for each modality), while OBS-Diff processes the two output projection matrices after separation. To resolve this, we fuse these rankings into a single, decisive list using Reciprocal Rank Fusion (RRF):

$$S_j^{\text{RRF}} = \frac{1}{k + \text{rank}_A(j)} + \frac{1}{k + \text{rank}_B(j)}, \tag{8}$$

where $\text{rank}_A(j)$ is the rank of head $j$ for modality A, and $k$ is a stabilizing hyperparameter (e.g., 60). This fused score provides a unified ranking to guide the pruning of shared attention heads.

Subsequently, the weights of the entire output projection layer are updated using the full Hessian matrix, $\mathbf{H}$, following the formulation presented in Eq. (6).

## 5 EXPERIMENTS

### 5.1 SETTINGS

**Models.** To demonstrate the generalizability of OBS-Diff, we evaluate it across a diverse range of text-to-image models: Stable Diffusion v2.1-base (866M) (Rombach et al., 2022), Stable Diffusion 3-Medium (2B) (Esser et al., 2024), Stable Diffusion 3.5-Large (8B), and Flux.1-dev (12B) (Black Forest Labs, 2024). For comparison with prior work, we also evaluate our method on DDPM (35.7M) (Ho et al., 2020) trained on the CIFAR-10 (32 × 32) dataset (Krizhevsky et al., 2009).

**Baselines.** For text-to-image models, we compare against methods adapted from the Large Language Model (LLM) domain for unstructured/semi-structured sparsity, namely Wanda (Sun et al., 2024) and DSnoT (Zhang et al., 2024d), as well as standard magnitude pruning. For structured pruning, we employ an L1-norm based baseline (Li et al., 2017) and EcoDiff (Zhang et al., 2024b). On

Table 1: Quantitative comparison of unstructured pruning methods on text-to-image diffusion models. The best result per metric is highlighted in **bold**.

(a) SD v2.1-base and SD 3-medium

| Base Model | Sparsity (%) | Method | FID ↓ | CLIP ↑ | ImageReward ↑ |
|---|---|---|---|---|---|
| | **Dense Model** | | 31.25 | 0.3142 | 0.3627 |
| SD v2.1-base | 40 | Magnitude | **27.86** | 0.3111 | 0.1864 |
| | | DSnoT | 31.63 | 0.3099 | -0.0422 |
| | | Wanda | 27.96 | 0.3122 | 0.1367 |
| | | OBS-Diff | 28.19 | **0.3131** | **0.2061** |
| | 50 | Magnitude | 49.38 | 0.2959 | -0.5580 |
| | | DSnoT | 69.05 | 0.2829 | -1.1395 |
| | | Wanda | 41.84 | 0.2988 | -0.4704 |
| | | OBS-Diff | **27.41** | **0.3102** | **-0.0356** |
| | **Dense Model** | | 36.14 | 0.3162 | 0.9029 |
| SD 3-medium | 50 | Magnitude | 221.24 | 0.1864 | -2.2719 |
| | | DSnoT | 63.37 | 0.2908 | -0.5941 |
| | | Wanda | 43.98 | 0.3000 | -0.1076 |
| | | OBS-Diff | **27.20** | **0.3167** | **0.6468** |
| | 60 | Magnitude | 349.53 | 0.1864 | -2.2807 |
| | | DSnoT | 211.58 | 0.2222 | -2.2271 |
| | | Wanda | 170.33 | 0.2352 | -2.0641 |
| | | OBS-Diff | **28.49** | **0.3099** | **0.1213** |

(b) SD 3.5-large and Flux 1.dev

| Base Model | Sparsity (%) | Method | FID ↓ | CLIP ↑ | ImageReward ↑ |
|---|---|---|---|---|---|
| | **Dense Model** | | 31.59 | 0.3156 | 0.7549 |
| SD 3.5-large | 50 | Magnitude | 35.21 | 0.3052 | 0.1465 |
| | | DSnoT | 32.82 | 0.3113 | 0.2323 |
| | | Wanda | **27.49** | 0.3123 | 0.4215 |
| | | OBS-Diff | 29.61 | **0.3142** | **0.6146** |
| | 60 | Magnitude | 156.21 | 0.2302 | -2.0296 |
| | | DSnoT | 81.99 | 0.2706 | -1.3198 |
| | | Wanda | 48.80 | 0.2859 | -0.6402 |
| | | OBS-Diff | **29.15** | **0.3119** | **0.3984** |
| | **Dense Model** | | 39.16 | 0.3110 | 0.9661 |
| Flux 1.dev | 60 | Magnitude | 42.06 | 0.2974 | -0.1945 |
| | | DSnoT | 41.55 | **0.3095** | 0.7111 |
| | | Wanda | **37.65** | 0.3086 | 0.7576 |
| | | OBS-Diff | 39.40 | 0.3075 | **0.7777** |
| | 70 | Magnitude | 251.58 | 0.2104 | -2.2271 |
| | | DSnoT | 44.35 | 0.2970 | -0.3459 |
| | | Wanda | 49.68 | 0.2957 | -0.1046 |
| | | OBS-Diff | **39.79** | **0.2986** | **0.3697** |

the CIFAR-10 DDPM, our method is directly compared with Diff-Pruning (Fang et al., 2023b). The sparsity refers to the pruning ratio of all the linear layers within MHA and FFN for each MMDiT block. For calibration, we utilize text prompts from the GCC3M dataset (Sharma et al., 2018). To ensure a fair comparison, all methods and baselines utilize identical configurations (computational resources provided in Appendix B).

The Wanda (Sun et al., 2024) and DSnoT (Zhang et al., 2024d) baselines are originally designed for unstructured and semi-structured pruning of LLMs. Their direct application to diffusion models is non-trivial due to the iterative nature of the diffusion model. Specifically, we extended their pruning logic by incorporating the concept of module packages, enabling them to perform unstructured and semi-structured pruning targeted at the key components of the diffusion architecture. Critically, to ensure an equitable comparison with our Hessian-based method, the adapted DSnoT baseline is configured to use its Hessian-based importance score calculation mode.

**Evaluation Metrics.** We evaluate the performance of the text-to-image models on a subset of 5K prompts from the MS-COCO 2014 validation set (Lin et al., 2014). The evaluation is based on three metrics: Fréchet Inception Distance (FID) (Heusel et al., 2017), CLIP Score (ViT-B/16) (Hessel et al., 2021), and ImageReward (Xu et al., 2023). For the DDPM on CIFAR-10, we report the FID score. We measure efficiency gains in terms of wall-clock time reduction and the decrease in FLOPs.

## 5.2 RESULTS OF UNSTRUCTURED PRUNING

The results in Table 1 show the superiority of our OBS-Diff in terms of CLIP score and ImageReward. An interesting phenomenon is observed with the FID metric – the pruned model can occasionally outperform the original dense model. *E.g.*, at 40% sparsity on SD v2.1-base, the *Magnitude* method beats the dense model in FID, while our results suggest *Magnitude* does not produce *visually* better results. It is thereby conceived that FID may not be a very reliable metric here to evaluate different pruning methods.

Table 2: Performance of semi-structured (2:4 sparsity pattern) pruning on the Stable Diffusion 3.5-Large model. Pruning is applied to the 3rd through 25th MMDiT blocks. The best result is shown in **bold**.

| Base Model | Method | FID ↓ | CLIP ↑ | ImageReward ↑ |
|---|---|---|---|---|
| | **Dense Model** | 31.59 | 0.3156 | 0.7549 |
| SD 3.5-Large | Magnitude | 45.39 | 0.2945 | -0.4705 |
| | DSnoT | 32.40 | 0.3069 | 0.0307 |
| | Wanda | **32.08** | 0.3036 | -0.1363 |
| | OBS-Diff | 32.13 | **0.3129** | **0.4493** |

Regarding the CLIP score, OBS-Diff is the best-performing method in the vast majority of test cases, exhibiting only a slight decrease compared to the dense models. Most notably, OBS-Diff consistently leads in the ImageReward metric across all benchmarks, indicating superior alignment with human aesthetic preferences.

Table 3: Performance of structured pruning on the **SDXL (U-Net)** model across various sparsity levels. Comparison includes the L1-norm baseline, EcoDiff, and our proposed OBS-Diff. The TFLOPs metric represents the theoretical computational cost for a single forward pass of the entire UNet. For each sparsity group, the best result per metric is highlighted in **bold**.

| Model | Sparsity | Method | #Params | TFLOPs ↓ | FID ↓ | CLIP Score ↑ | ImageReward ↑ |
|-------|----------|--------|---------|----------|-------|--------------|---------------|
| | **Dense Model** | | 2.57 B | 5.98 | 29.21 | 0.3213 | 0.7635 |
| | 15% | $L_1$-norm | | | 71.78 | 0.3035 | -0.0006 |
| | | EcoDiff | 2.24 B | 5.33 (↓10.87%) | 34.18 | 0.3100 | -0.1870 |
| | | **OBS-Diff (Ours)** | | | **29.08** | **0.3215** | **0.6877** |
| SDXL | 20% | $L_1$-norm | | | 133.07 | 0.2825 | -0.7897 |
| | | EcoDiff | 2.13 B | 5.12 (↓14.38%) | 42.98 | 0.2993 | -0.6172 |
| | | **OBS-Diff (Ours)** | | | **29.19** | **0.3212** | **0.6461** |
| | 30% | $L_1$-norm | | | 170.68 | 0.2711 | -1.1694 |
| | | EcoDiff | 1.91 B | 4.70 (↓21.40%) | 101.96 | 0.2465 | -1.9161 |
| | | **OBS-Diff (Ours)** | | | **29.75** | **0.3204** | **0.4909** |

The superiority of our approach becomes most pronounced at high sparsity levels. For example, at 60% sparsity on SD 3.5-Large or 70% on Flux 1.dev, the performance of all baseline methods collapses, resulting in metrics that are significantly worse than ours. This quantitative degradation corresponds to a qualitative failure; as illustrated in Figure 1, the images generated by baseline methods at high sparsity are often totally destroyed and suffer from severe artifacts, whereas OBS-Diff continues to produce high-quality and coherent results. Beyond its performance in generation quality, OBS-Diff is also highly efficient. For instance, the entire pruning process for the 2B-parameter SD 3-medium model completes in under 15 minutes on a single NVIDIA RTX 4090, highlighting its excellent cost-effectiveness. Detailed analyses of pruning time and the impact of sparsity on ImageReward are provided in Appendix C.1.

## 5.3 RESULTS ON SEMI-STRUCTURED PRUNING

The results for 2:4 semi-structured pruning are presented in Table 2. Although Wanda obtains a slightly better FID of 32.08 compared to our 32.13, OBS-Diff shows substantial advantages in semantic-level metrics. Notably, it surpasses the strongest baseline by a large margin in both CLIP score (0.3129) and ImageReward (0.4493). This highlights our method's effectiveness in maintaining high-level semantic consistency and visual fidelity under hardware-friendly sparsity constraints.

## 5.4 RESULTS ON STRUCTURED PRUNING

The results are presented in Table 4 and Table 3. The baseline $L_1$-norm pruning suffers from catastrophic performance degradation even at a modest 15% sparsity, with its FID score deteriorating from 31.59 to 158.89 on SD 3.5-Large. In stark contrast, our method, OBS-Diff, demonstrates remarkable resilience. At the same 15% sparsity, OBS-Diff maintains an FID of 32.64, nearly identical to the dense model's performance. This robustness persists up to 30% sparsity, where OBS-Diff sustains a strong FID of 34.51 while the baseline model fails completely (FID of 327.48). These findings highlight OBS-Diff's superior ability to preserve critical model structures under aggressive structured pruning.

To benchmark our method against established techniques, we incorporate the comparison with EcoDiff (Zhang et al., 2024b), a state-of-the-art structured pruning framework for text-to-image diffusion models, directly into the main experiments. As shown in Table 3 and Table 4, while EcoDiff generally outperforms the naive $L_1$-norm baseline on SDXL, it still exhibits significant performance degradation compared to our method on both tables, especially at higher sparsity levels. For instance, on the U-Net based SDXL model (Table 3), EcoDiff yields an FID of 101.96 at 30% sparsity, whereas OBS-Diff achieves a substantially better FID of 29.75. This confirms that OBS-Diff generalizes effectively across diverse architectures, outperforming baselines on both MMDiT (SD 3.5) and U-Net (SDXL) backbones.

Finally, we compare OBS-Diff with Diff-Pruning (Fang et al., 2023b), a well-recognized method that leverages gradient information for structured pruning on small class-conditional DDPMs. The

Table 4: Performance of structured pruning on the Stable Diffusion 3.5-Large model across various sparsity levels. The first and last transformer blocks were excluded from the pruning process. The TFLOPs metric represents the theoretical computational cost for a single forward pass of the entire transformer. For each sparsity group, the best result per metric is highlighted in **bold**.

| Base Model | Sparsity (%) | Method | #Params | TFLOPs ↓ | FID ↓ | CLIP ↑ | ImageReward ↑ |
|---|---|---|---|---|---|---|---|
| | **Dense Model** | | 8.06 B | 11.26 | 31.59 | 0.3156 | 0.7549 |
| | 15% | $L_1$-norm | 7.28 B | 9.63 (↓14.5%) | 158.89 | 0.2376 | -2.0502 |
| | | EcoDiff | | | 230.97 | 0.2086 | -2.2594 |
| | | OBS-Diff | | | **32.64** | **0.3157** | **0.6446** |
| | 20% | $L_1$-norm | 7.02 B | 9.09 (↓19.3%) | 189.50 | 0.2124 | -2.2385 |
| | | EcoDiff | | | 293.89 | 0.2050 | -2.2724 |
| SD 3.5-Large | | OBS-Diff | | | **32.46** | **0.3149** | **0.5475** |
| | 25% | $L_1$-norm | 6.76 B | 8.55 (↓24.1%) | 228.82 | 0.2040 | -2.2651 |
| | | EcoDiff | | | 308.96 | 0.2037 | -2.2686 |
| | | OBS-Diff | | | **33.73** | **0.3128** | **0.3741** |
| | 30% | $L_1$-norm | 6.54 B | 8.10 (↓28.1%) | 327.48 | 0.2093 | -2.2663 |
| | | EcoDiff | | | 346.38 | 0.2024 | -2.2746 |
| | | OBS-Diff | | | **34.51** | **0.3107** | **0.2221** |

Table 6: Ablation study of timestep weighting strategies, conducted on the SD3-Medium model at 50% unstructured sparsity. (For reference, the ImageReward of uniform strategy is 0.6355.)

| Weight strategy | ImageReward ↑ |
|---|---|
| Linear increase | 0.6174 |
| Linear decrease | 0.6384 |
| Log increase | 0.6244 |
| **Log decrease** | **0.6438** |

Table 7: Ablation study on the impact of the number of module packages on resource usage and performance, conducted on SD3-Medium model at 30% unstructured sparsity.

| Pkgs. | Mem. (GB)↓ | Time (s)↓ | ImageReward↑ |
|---|---|---|---|
| 1 | 30.67 | 572.20 | 0.8569 |
| 4 | 24.05 | 896.52 | 0.8442 |
| 10 | 22.75 | 1539.37 | 0.8429 |
| 20 | 22.08 | 2594.95 | 0.8564 |

detailed results for this specific comparison are deferred to the Appendix C.3, where our method outperforms Diff-Pruning consistently.

## 5.5 WALL-CLOCK TIME COMPARISON

To quantify the practical efficiency gains, we measure the wall-clock time for a single forward pass through an MMDiT block of the SD3.5-Large model, on a single NVIDIA 4090 GPU with batch size 4, resolution 1024×1024.

Table 5 shows that both methods effectively reduce inference latency. The 2:4 semi-structured approach achieves a 1.23× speedup, while our structured pruning method attains 1.31× speedup at 30% sparsity. These results validate the tangible practical acceleration benefits of applying these pruning techniques.

Table 5: Wall-clock inference time (ms) and speedup for a single MMDiT block under various sparsity schemes.

| Sparsity Type | Time (ms) | Speedup |
|---|---|---|
| Dense | 14.36 | / |
| Semi-structured (2:4) | 11.71 | 1.23× |
| Structured (15%) | 13.96 | 1.03× |
| Structured (20%) | 11.95 | 1.20× |
| Structured (25%) | 11.17 | 1.29× |
| Structured (30%) | 10.99 | 1.31× |

## 5.6 ABLATION STUDY

We perform an ablation study to analyze the impact of three key components: (1) the timestep-aware Hessian construction, (2) the number of module packages, and (3) the number of prompts in the calibration dataset. For this study, all variants are evaluated using the ImageReward metric on 1,000 prompts from the MS-COCO 2014 validation set.

**Timestep-Aware Hessian Matrix Establishment.** To incorporate temporal information from the diffusion process, we introduce timestep-aware weighting during the Hessian matrix construction. This method assigns a distinct weight to the hooked activations at each timestep. Empirical results demonstrate that assigning greater importance to earlier inference steps yields superior performance. As shown in Table 6, a logarithmic decrease strategy significantly outperforms other weight distribution methods.

**Module-Package.** The concept of module packages partitions the model's layers for layer-wise compression. This approach introduces a critical trade-off between computational resources and time. Processing the model in more packages reduces peak GPU memory, as the Hessian matrix for each pruning step is smaller. However, it proportionally increases the total runtime because the entire calibration dataset must be forwarded for each package. As shown in our ablation study (Table 7), while the resource trade-off is evident, the number of packages does not show a clear, predictable relationship with the final pruned model's performance. Consequently, practitioners can select a configuration that best fits their hardware constraints without sacrificing final model quality.

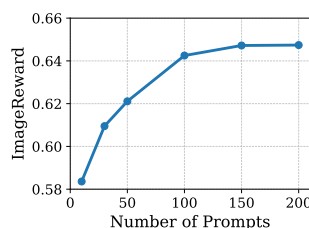

Figure 3: Effect of the number of prompts in calibration dataset on the ImageReward.

**The Number of the Prompts in the Calibration Dataset.** The size of the calibration dataset is a critical hyperparameter that directly influences the quality of the approximated Hessian matrix. To find an optimal size, we evaluated post-pruning performance against the number of text prompts in the calibration dataset, as shown in Figure 3. The pruned model's ImageReward score improves sharply up to 100 prompts and then plateaus, indicating a point of diminishing returns where additional data offers no significant benefit to the Hessian approximation. Therefore, to balance performance gains with computational efficiency, we selected 100 prompts for our calibration dataset in all main experiments.

## 6 CONCLUSION

This work introduces OBS-Diff, a novel one-shot, training-free pruning framework tailored for large-scale text-to-image diffusion models. By revitalizing the classic Optimal Brain Surgeon method, we address the unique challenges of iterative denoising through our proposed timestep-aware Hessian construction, which prioritizes critical early-stage generation steps. To overcome prohibitive calibration costs, we devise a group-wise sequential pruning strategy that effectively balances memory overhead and computational efficiency. The versatility of our framework extends across unstructured, semi-structured, and structured pruning, demonstrating its broad applicability. Extensive empirical results show that OBS-Diff establishes a new state-of-the-art in training-free diffusion model pruning, consistently outperforming existing methods by maintaining high generative quality, especially at high sparsity regimes.

## ACKNOWLEDGEMENT

This paper is supported by Young Scientists Fund of the National Natural Science Foundation of China (NSFC) (No. 62506305), Zhejiang Leading Innovative and Entrepreneur Team Introduction Program (No. 2024R01007), Key Research and Development Program of Zhejiang Province (No. 2025C01026), Scientific Research Project of Westlake University (No. WU2025WF003), Chinese Association for Artificial Intelligence (CAAI) & Ant Group Research Fund - AGI Track (No. 2025CAAI-ANT-13). It is also supported by the research funds of the National Talent Program and Hangzhou Municipal Talent Program.

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

## A   DECLARATION OF LLM USAGE

The use of Large Language Models (LLMs) in this work served two purposes: (1) to aid and polish the paper writing, and (2) to generate some of the text prompts used by the diffusion model to create figures that are shown in the paper.

## B   IMPLEMENTATION DETAILS

This section provides further details on the experimental setup, including common configurations, baseline adaptations, and the computational hardware used for our evaluations.

**Common Configurations.**   To ensure a controlled and fair comparison, all experiments, unless otherwise specified, adhere to a common set of configurations. For all text-to-image generation tasks, we set the output resolution to $512 \times 512$ pixels to facilitate rapid experimentation across the diverse and large-scale models. For our method and baselines such as Wanda (Sun et al., 2024) and DSnoT (Zhang et al., 2024d), we consistently group model parameters into 4 module packages. Furthermore, a logarithmic decreasing timestep weighting scheme (log decrease) was uniformly applied across all diffusion models and pruning methods to schedule the pruning process over the diffusion timesteps.

**Computational Resources.**   The training of the DDPM on the CIFAR-10 dataset was conducted on NVIDIA A100 GPUs. For the large text-to-image models, all pruning methods are training-free. The pruning and evaluation for Stable Diffusion v2.1-base, Stable Diffusion 3-Medium, and Stable Diffusion 3.5-Large were performed on a single NVIDIA RTX 4090 GPU, each equipped with 48GB of VRAM. Due to its substantial memory footprint, all experiments involving the FLUX.1-dev model, including its pruning and evaluation, was conducted on a single NVIDIA A100 GPU with 80GB of VRAM.

## C   MORE EXPERIMENTAL RESULTS

### C.1   MORE ANALYSIS FOR UNSTRUCTUREDLY PRUNED SD3-MEDIUM

As illustrated in Figure 4, our proposed OBS-Diff method consistently outperforms all baseline approaches in terms of the ImageReward metric across all evaluated sparsity levels. The superiority of our method is particularly pronounced at higher sparsity ratios. For instance, at 60% sparsity, the performance of competing methods collapses, yielding negative ImageReward scores. In stark contrast, OBS-Diff maintains a positive score, demonstrating its exceptional robustness in high-compression scenarios.

In terms of computational efficiency, Table 8 indicates that OBS-Diff has the longest pruning time on a single NVIDIA RTX 4090. However, the additional overhead is marginal, requiring only slightly more time than DSnoT (14.95 vs. 14.25 minutes). Considering the substantial gains in generation quality and model robustness, we conclude that OBS-Diff offers a superior trade-off between performance and computational cost, establishing it as a highly cost-effective pruning solution.

Table 8: Pruning time of different unstructured pruning methods on SD3-Medium (2B) at 50% sparsity.

| Method | Time (min) |
|---|---|
| Magnitude | $\approx 0$ |
| Wanda | 7.32 |
| DSnoT | 14.25 |
| OBS-Diff | 14.95 |

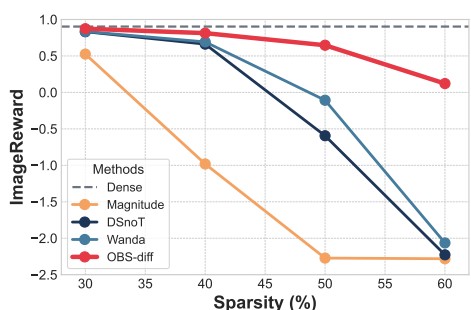

Figure 4: ImageReward vs. sparsity for various unstructured pruning methods on SD3-Medium.

## C.2 STRUCTURED PRUNING FOR SD3-MEDIUM

We evaluate our structured pruning method, OBS-diff, on the Stable Diffusion 3-medium model and compare it against the widely-used $L_1$-norm magnitude pruning baseline. As summarized in Table 9, the baseline method suffers from severe performance degradation as sparsity increases. In contrast, our approach maintains performance remarkably close to the original dense model across all tested sparsity levels, demonstrating its effectiveness and robustness.

Table 9: Performance comparison of structured pruning methods at 10%, 15%, 20%, and 25% sparsity on the Stable Diffusion 3-medium model (2B). The first and last transformer blocks were excluded from the pruning process. The TFLOPs metric represents the theoretical computational cost for a single forward pass of the entire transformer. For each sparsity group, the best result per metric is highlighted in **bold**.

| Base Model | Sparsity (%) | Method | #Params | TFLOPs ↓ | FID ↓ | CLIP ↑ | ImageReward ↑ |
|---|---|---|---|---|---|---|---|
| | **Dense Model** | | 2.03 B | 2.84 | 36.14 | 0.3162 | 0.9029 |
| | 10% | $L_1 - norm$
Ours (OBS-diff) | 1.91 B | 2.59 (↓8.8%) | 267.32
**35.65** | 0.2035
**0.3166** | -2.2611
**0.8118** |
| | 15% | $L_1 - norm$
Ours (OBS-diff) | 1.83 B | 2.43 (↓14.4%) | 326.92
**34.33** | 0.1942
**0.3168** | -2.2768
**0.6717** |
| SD 3-medium | 20% | $L_1 - norm$
Ours (OBS-diff) | 1.78 B | 2.31 (↓18.7%) | 348.77
**33.15** | 0.1926
**0.3163** | -2.2768
**0.4997** |
| | 25% | $L_1 - norm$
Ours (OBS-diff) | 1.72 B | 2.19 (↓22.9%) | 365.24
**32.96** | 0.1906
**0.3143** | -2.2786
**0.2782** |

## C.3 COMPARISON WITH DIFF-PRUNING ON DDPM

To evaluate the generalizability of our method beyond large-scale text-to-image models, we adapt it to the task of structured pruning for a Denoising Diffusion Probabilistic Model (DDPM) on the CIFAR-10 dataset. Our adaptation leverages the column masks identified by Diff-Pruning, which are then integrated with our OBS weight update mechanism as detailed in Eq. (6).

As presented in Table 10, our method surpasses the current state-of-the-art baseline, Diff-Pruning, by achieving a superior FID score under an identical fine-tuning budget (100K steps). This result demonstrates not only the versatility of our approach but also suggests that the model pruned by OBS-Diff serves as a more effective checkpoint for subsequent fine-tuning.

Table 10: **Performance of pruned DDPMs on CIFAR-10** ($32 \times 32$). All pruned models are fine-tuned for 100K steps. Evaluations are conducted on samples generated via 100 DDIM steps. The best FID score is highlighted in **bold**.

| Method | #Params ↓ | MACs ↓ | FID ↓ | Train Steps ↓ |
|---|---|---|---|---|
| Pretrained | 35.7M | 6.1G | 4.50 | 800K |
| Random Pruning | 13.95M | 2.1G | 7.85 | 100K |
| Magnitude Pruning | 13.95M | 2.1G | 7.91 | 100K |
| Diff-Pruning | 13.95M | 2.1G | 7.72 | 100K |
| **OBS-Diff** | 13.95M | 2.1G | **7.55** | 100K |

## D  ROBUSTNESS AND GENERALIZATION ANALYSIS

To demonstrate that our calibration (using only 100 prompts) does not overfit, we evaluated the fixed pruned model (**SD3-Medium, 50% Unstructured**) under inference conditions significantly different from the calibration settings (CFG 7.0, Steps 25, Euler). For fast evaluation, we evaluated all tasks on the MSCOCO 2014 validation 1K subset using the ImageReward metric.

### D.1  ROBUSTNESS TO CFG SCALES

We evaluated the pruned model across varying Classifier-Free Guidance (CFG) scales. As shown in Table 11, the pruned model achieves **higher performance at CFG 9.0 (0.7044)** than at the calibration setting of CFG 7.0 (0.6425). This indicates that our pruning strategy effectively preserves the model's semantic generation capabilities and generalizes exceptionally well to higher guidance scales, which are critical for high-quality text-to-image synthesis.

Table 11: Robustness of the pruned SD3-Medium (50% Unstructured) across different CFG scales. The model was calibrated at CFG 7.0.

| CFG | Dense (ImageReward) | Pruned (ImageReward) | Performance |
|---|---|---|---|
| 5.0 | 0.8319 | 0.5297 | - |
| 7.0 (Calibrated) | 0.8510 | 0.6425 | Baseline |
| 9.0 | 0.8275 | **0.7044** | **Improved** |

### D.2  ROBUSTNESS TO SAMPLING STEPS

To address the concern regarding step counts, we evaluated the pruned model (calibrated at 25 steps) across 15, 25, and 50 inference steps. The results are summarized in Table 12. Although calibrated at 25 steps, the pruned model effectively leverages additional compute at 50 steps to generate higher-quality images. This confirms the pruning preserves the integrity of the underlying ODE trajectory.

Table 12: Robustness across varying inference sampling steps. The model was calibrated at 25 steps.

| Steps | Dense (ImageReward) | Pruned (ImageReward) | Trend |
|---|---|---|---|
| 15 | 0.6988 | 0.4883 | Fast Preview |
| 25 (Calibrated) | 0.8510 | 0.6425 | Baseline |
| 50 | **0.9391** | **0.7153** | **Improved Quality** |

### D.3  ROBUSTNESS ACROSS SAMPLERS

We evaluated generalization across different solvers on both SD3-Medium (MMDiT) and SD v2.1 (U-Net).

- **SD3-Medium:** Calibrated on Euler (1st-order), the model generalizes zero-shot to Heun (2nd-order), showing significant quality gains.

- **SD v2.1:** We applied 40% unstructured pruning (calibrated on PNDM). As shown in Table 13, the relative performance ranking of the samplers is preserved between the Dense and Pruned models (e.g., DPM++ remains the highest performing), indicating the pruning is solver-agnostic.

Table 13: Generalization across different samplers for SD3-Medium and SD v2.1.

| Model | Sampler | ImageReward (Dense) | ImageReward (Pruned) |
|---|---|---|---|
| **SD3-Medium** | Euler (Calibrated) | 0.8510 | 0.6425 |
| | Heun (2nd Order) | **0.9200** | **0.7249** |
| **SD v2.1** | PNDM (Calibrated) | 0.3432 | 0.1782 |
| | DPM++ | **0.3889** | **0.2246** |
| | EDM | 0.3442 | 0.1534 |
| | DDIM | 0.3439 | 0.1579 |

### D.4 GENERALIZATION TO OUT-OF-DISTRIBUTION (OOD) PROMPTS

We address the concern regarding calibration data in two ways:

1. **Experimental Design:** All main results in the paper already use **GCC3M for calibration** and **MS-COCO for evaluation**, representing a standard OOD setting.

2. **New Validation Experiment:** To rigorously test this, we calibrated two separate models—**one using MS-COCO 2014 Train (In-Distribution)** and **one using GCC3M (Out-of-Distribution)**—and evaluated both on the **MS-COCO 2014 validation 5K subset**.

As shown in Table 14, the performance is nearly identical. The model calibrated on OOD data (GCC3M) performs on par with (and even slightly better in FID than) the ID model. This definitively proves that OBS-Diff captures generalizable features and does not overfit to the calibration prompts.

Table 14: Comparison of models calibrated on In-Distribution (MS-COCO) vs. Out-of-Distribution (GCC3M) datasets, evaluated on MS-COCO validation set.

| Calibration Dataset | Evaluation | FID ↓ | CLIP Score ↑ | ImageReward ↑ |
|---|---|---|---|---|
| MS-COCO 2014 Train | In-Distribution (ID) | 27.93 | 0.3169 | 0.6547 |
| GCC3M Train | Out-of-Distribution (OOD) | 27.20 | 0.3167 | 0.6468 |

## E ADDITIONAL QUALITATIVE RESULTS

In this section, we provide additional qualitative results to further demonstrate the effectiveness of OBS-Diff. Figures 5 to 7 and Figure 8 showcase unstructured pruning comparisons on SD3-Medium and Flux 1.dev, respectively, benchmarking against Magnitude, DSnoT, and Wanda. Furthermore, Figures 9 to 11 illustrate structured pruning results on SD3.5-Large compared to the L1-norm baseline. Across these diverse architectures and high sparsity regimes, OBS-Diff consistently maintains superior visual fidelity and semantic consistency compared to existing methods.

## F FUTURE WORK

While OBS-Diff demonstrates compelling effectiveness in training-free, one-shot pruning for text-to-image diffusion models, its potential extends far beyond this scope. We envision three strategic directions to broaden its impact:

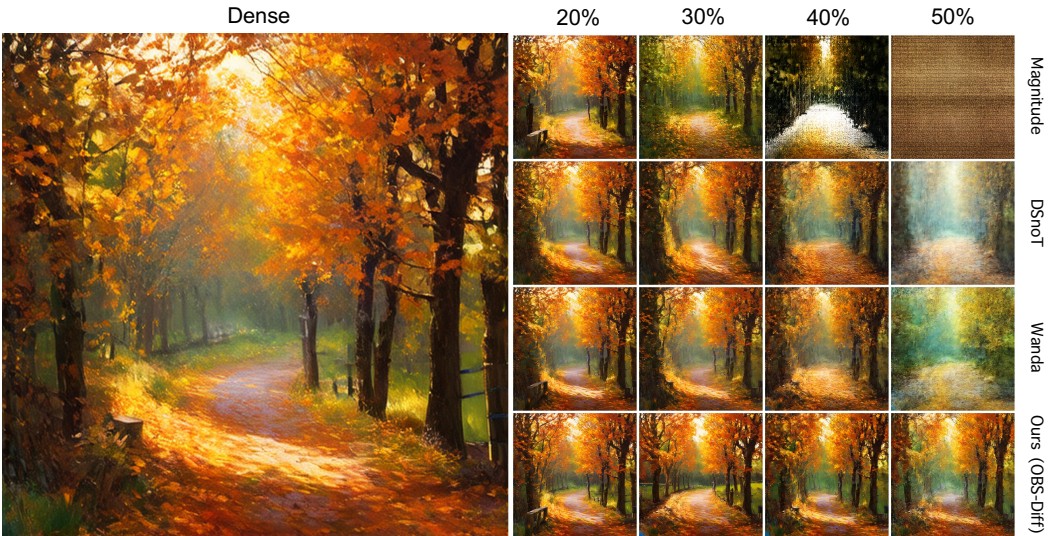

Figure 5: More Qualitative comparison of unstructured pruning methods on the SD3-Medium model. We evaluate Magnitude, DSnoT, Wanda, and our method (OBS-Diff) at various sparsity levels (20%, 30%, 40%, and 50%) using the same prompt and negative prompt. All images are generated at a resolution of $512 \times 512$.

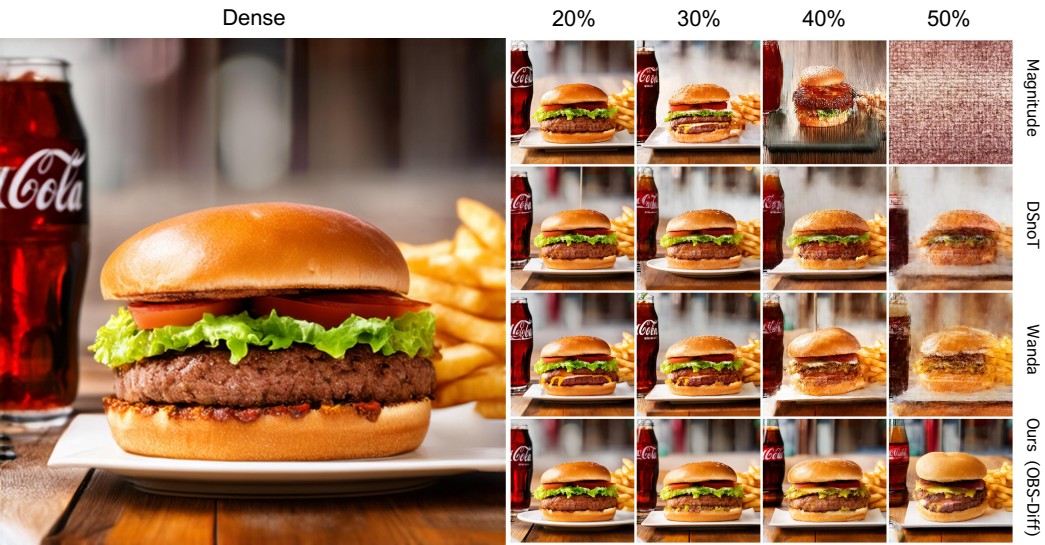

Figure 6: More Qualitative comparison of unstructured pruning methods on the SD3-Medium model. We evaluate Magnitude, DSnoT, Wanda, and our method (OBS-Diff) at various sparsity levels (20%, 30%, 40%, and 50%) using the same prompt and negative prompt. All images are generated at a resolution of $512 \times 512$.

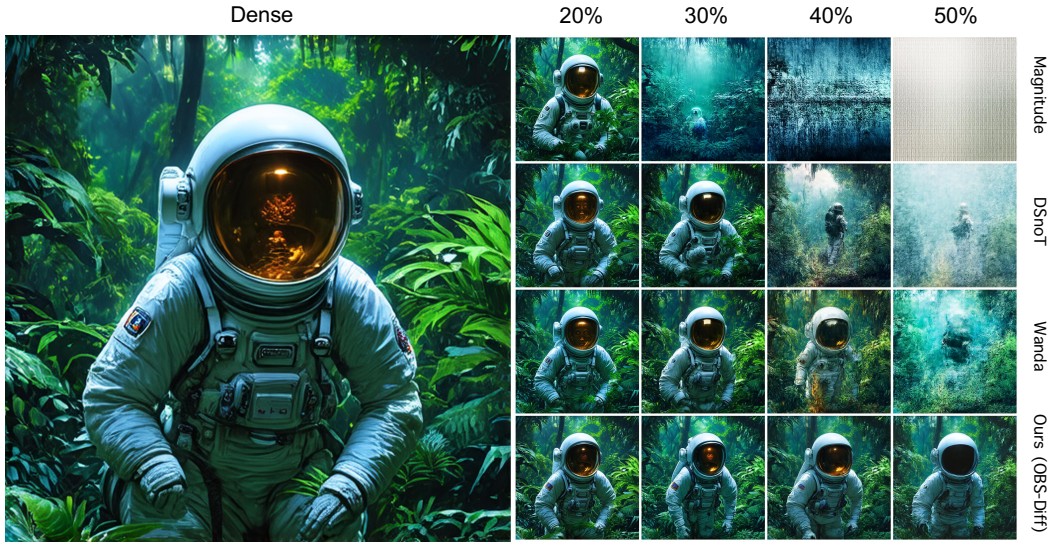

Figure 7: More Qualitative comparison of unstructured pruning methods on the SD3-Medium model. We evaluate Magnitude, DSnoT, Wanda, and our method (OBS-Diff) at various sparsity levels (20%, 30%, 40%, and 50%) using the same prompt and negative prompt. All images are generated at a resolution of $512 \times 512$.

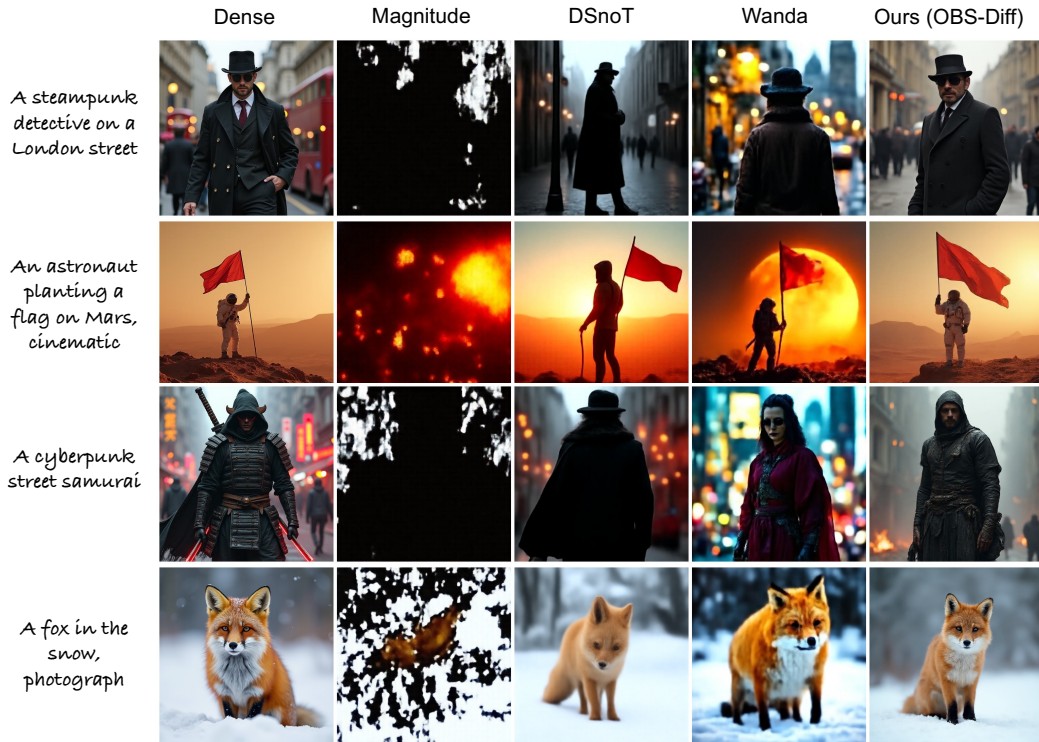

Figure 8: Qualitative comparison of unstructured pruning methods on Flux 1.dev at 70% sparsity. Results from Magnitude, DSnoT, Wanda, and our proposed OBS-Diff are shown.

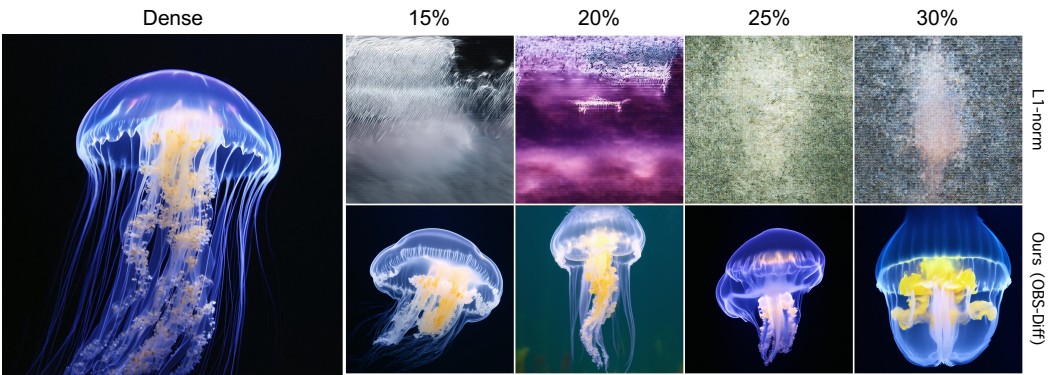

Figure 9: Qualitative comparison of structured pruning methods on the SD3.5-Large model at various sparsity levels (15%, 20%, 25%, and 30%). Results from the L1-norm baseline and our proposed OBS-Diff are shown.

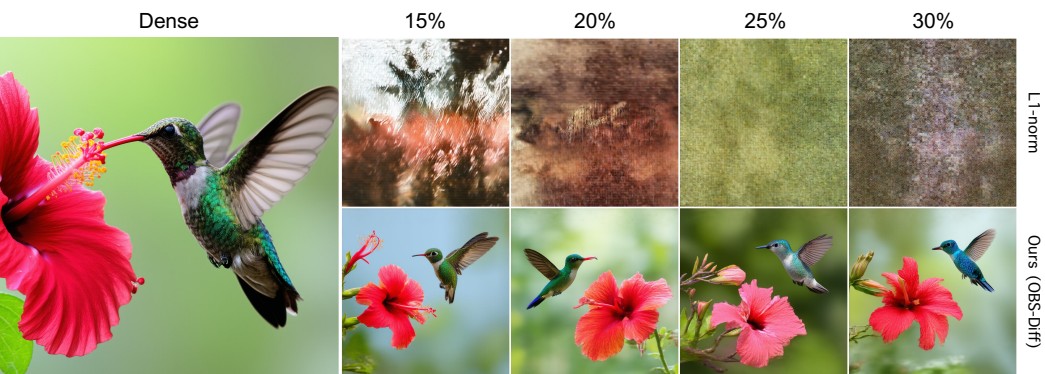

Figure 10: Qualitative comparison of structured pruning methods on the SD3.5-Large model at various sparsity levels (15%, 20%, 25%, and 30%). Results from the L1-norm baseline and our proposed OBS-Diff are shown.

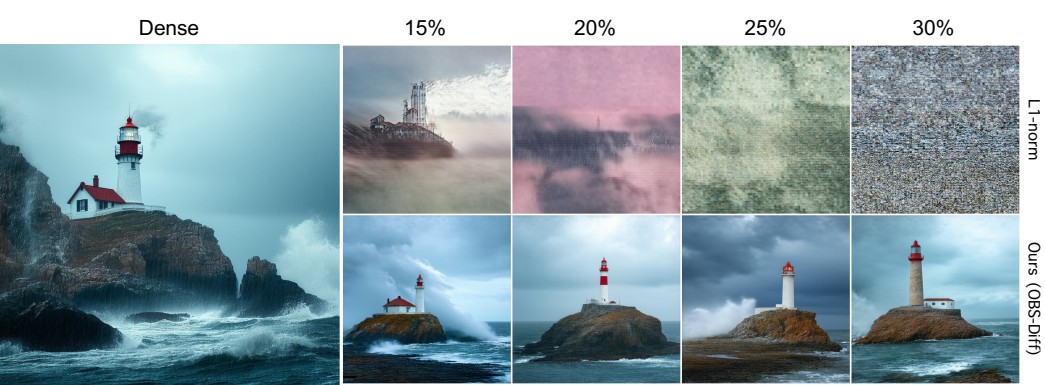

Figure 11: Qualitative comparison of structured pruning methods on the SD3.5-Large model at various sparsity levels (15%, 20%, 25%, and 30%). Results from the L1-norm baseline and our proposed OBS-Diff are shown.

**Generalization to diverse diffusion paradigms.** OBS-Diff revitalizes the classical Optimal Brain Surgeon framework through elegant designs tailored for the iterative denoising process. This theoretical foundation is transferable to broader diffusion-based architectures, such as Diffusion LLMs (dLLMs) (Nie et al., 2025; Ye et al., 2025; Zhu et al., 2025; Chen et al., 2025b), video generation models (Kong et al., 2024; Wan et al., 2025), and other diffusion-based models (Cao et al., 2025; Abramson et al., 2024; Wang et al., 2026).

**Extension to other architectures.** OBS-Diff revitalizes the traditional Optimal Brain Surgeon pruning algorithm for diffusion models by incorporating designs specifically tailored to the iterative nature of the process. The potential of this Hessian-based pruning paradigm can be further applied to recent advanced architectures, such as Vision-Language Models (VLMs) (Zhang et al., 2023; Li et al., 2024; Zhang et al., 2024c; Diao et al., 2025a; 2024; 2025b), reasoning models (Feng et al., 2025c;b; Du et al., 2025; Fang et al., 2025b; Chen et al., 2025c; Feng et al., 2025a), and unified multimodal models (Tian et al., 2025; Lu et al., 2025; Deng et al., 2025), to explore inherent sparsity and enable efficient inference.

**Synergy with other efficiency techniques.** To further enhance model efficiency, OBS-Diff and its possible extensions serve as weight pruning methods that are orthogonal to other acceleration techniques. It can be combined with approaches such as Token Merging (Tao et al., 2025; Shao et al., 2025; Chen et al., 2025a; Jin et al., 2025; Yang et al., 2025), sparse attention (Zhang et al., 2025; Li et al., 2026), or other compression methods (Bai et al., 2025; Li et al., 2025) to achieve even greater efficiency gains.

