# OpenReview forum: "OBS-Diff: Accurate Pruning For Diffusion Models in One-Shot"
_ICLR.cc/2026/Conference — ICLR 2026 Poster_

### Official Review · Reviewer_ePsz · 2025-10-31

**Soundness:** 3
**Presentation:** 3
**Contribution:** 3
**Rating:** 6
**Confidence:** 3

**Summary:**

This paper proposes a **training-free OBS-Diff framework** for pruning multi-step diffusion models to reduce computation cost. It extends the **Optimal Brain Surgeon (OBS)** algorithm to diffusion models, using a **timestep-aware Hessian** to reflect the varying importance of parameters across denoising steps. The authors also design **Module Packages** to group layers for batch pruning, reducing the cost of activation collection. OBS-Diff supports **unstructured**, **semi-structured**, and **structured** pruning. Experiments show that it outperforms existing methods in ImageReward and CLIP score. And the quantitative experiment shows that it perform much better than other methods at 30% and 40% sparsity level.

**Strengths:**

1. The method is **highly generalizable**, applicable to different diffusion architectures such as U-Net and MMDiT.
2. It **extends OBS pruning** to diffusion models, introducing timestep-aware weighting that better preserves early-stage generation accuracy.
3. Supports multiple pruning granularities (unstructured, semi-structured, structured) in a unified framework.

**Weaknesses:**

1. The pruning ratio is fixed per layer (layer-wise), without dynamic or weighted adjustment based on layer importance. Different layers may contribute unequally to the final output, so uniform sparsity could be suboptimal.

**Questions:**

1. Why doesn’t the paper include a **comparison of inference-time acceleration** between different pruning methods? Only pruning-time cost is reported.
2. Is the **generalization ability limited by the calibration dataset**? Since pruning depends on calibration prompts, will prompts outside that distribution generate lower-quality results?
3. If the **calibration dataset size and prompt diversity** increase, could neuron activations become more uniform, reducing contrast in neuron importance and potentially leading to over-pruning of useful neurons?

---

> ### Author Response · Authors · 2025-11-23
> **Response to Reviewer ePsz (Part-1)**
>
> We sincerely thank the reviewer for the thoughtful feedback and for recognizing OBS-Diff as a "highly generalizable" framework that "better preserves early-stage generation accuracy" and successfully supports "multiple pruning granularities in a unified framework."
>
> We have uploaded a revised PDF containing new/revised data (highlighted in blue). We value your constructive suggestions and address your questions below.
>
> ---
>
> > **Weakness**: "Uniform vs. Dynamic Layer-wise Sparsity"
>
> We agree that dynamic sparsity allocation is a promising direction. However, we strictly adhered to uniform sparsity in this work for two critical reasons:
>
> 1. **Controlled Comparison:** To rigorously evaluate the effectiveness of the pruning metric itself (e.g., our Timestep-Aware Hessian vs. Wanda/Magnitude), we must keep the sparsity distribution constant. Using dynamic allocation would conflate the benefits of the allocation strategy with the pruning metric, making the comparison unfair.
>
> 2. **Baseline Establishment:** As the first work to enable accurate, training-free, one-shot pruning for large-scale diffusion models, our goal was to *establish a solid baseline.* We view dynamic allocation as an orthogonal optimization to be built on top of OBS-Diff in future work.
>
> > **Q1:** Comparison of Inference-time Acceleration
>
> We clarify that inference acceleration is determined solely by the final sparsity level and pattern, not the pruning algorithm used to select weights.
>
> + **Identical Speed:** In our experiments, all methods (e.g., OBS-Diff vs. L1-norm) are evaluated at the exact same sparsity (e.g., 30% structured). Therefore, they possess identical architecture, TFLOPs, and inference latency.
>
>
> + **Focus on Quality:** Since speed is constant across methods at a given sparsity, the only meaningful differentiator is generation quality (FID, ImageReward), which is the focus of our reporting.
>
>
> >**Q2:** Generalization & Calibration Dataset (OOD)
>
> Thank you for this critical question. Our experimental design directly addresses this concern:
>
> 1.  **Separate Datasets:** We explicitly used a different dataset for calibration than for evaluation. We used prompts from the **GCC3M dataset** for calibration.
> 2.  **OOD Evaluation:** All our main performance evaluations were conducted on a 5K subset of the **MS-COCO 2014 validation set**.
> 3.  Since the evaluation was performed on a dataset (MS-COCO) that is out-of-distribution relative to the calibration dataset (GCC3M), the strong results (e.g., in Table 1) already demonstrate that our method generalizes well and is not limited to the calibration distribution.
>
> To rigorously validate this, we further conducted a direct comparison using two distinct, large-scale calibration datasets. We evaluated the resulting pruned models on the *exact same* **MS-COCO 2014 validation 5k set**.
>
> 1.  **Calibration Set 1:** **MS-COCO 2014 training split** (acting as an "in-distribution" or ID set).
> 2.  **Calibration Set 2:** **GCC3M training split** (acting as an "out-of-distribution" or OOD set relative to the evaluation set).
>
> The results below show that the final model performance is nearly identical, regardless of which calibration set was used.
>
> | Calibration Dataset | Evaluation (on MS-COCO 5k) | FID $\downarrow$ | CLIP Score $\uparrow$ | ImageReward $\uparrow$ |
> | :--- | :--- | :--- | :--- | :--- |
> | MS-COCO 2014 Train | In-Distribution (ID) | 27.93 | 0.3169 | 0.6547 |
> | GCC3M Train | Out-of-Distribution (OOD) | 27.20 | 0.3167 | 0.6468 |
>
> >**Q3:** Calibration Size & Neuron Importance Contrast
>
> We thank the reviewer for this insightful theoretical question. The concern is that increasing calibration data diversity might "flatten" the importance scores, leading to a performance drop from over-pruning useful neurons.
>
>
> To specifically address this concern, we re-conduct the ablation study on the number of calibration prompts on 4090 GPU with module-package equaling 4 for each individual experiment and measured the post-pruning ImageReward for 1K prompts from the MSCOCO 2014 validation dataset.
>
>  The calibration datasets are randomly chosen from the GCC3M dataset. The results are as follows:
>
> | Number of Prompts | ImageReward (Post-Pruning) |
> | :--- | :--- |
> | 10 | 0.5836 |
> | 30 | 0.6095 |
> | 50 | 0.6211 |
> | 100 | 0.6425 |
> | 150 | 0.6472 |
> | 200 | 0.6474 |
>
> **Conclusion:** Contrary to the concern, adding data improves performance.
>
> 1. **Better Approximation:** More data leads to a more accurate Hessian approximation, reducing pruning error.
>
> 2. **Convergence:** Performance does not degrade; it improves and then plateaus/converges around 100-150 prompts. This confirms that useful neurons are correctly identified and preserved, rather than over-pruned.
>
> ---

---

> ### Author Response · Authors · 2025-11-23
> **Response to Reviewer ePsz (Part-2)**
>
> We hope these clarifications, specifically the evidence regarding OOD robustness and calibration convergence, can fully address your concerns. Given these substantiated improvements, we respectfully ask: Does the current version of our work address the limitations that previously held back a higher rating? We trust in your fair judgment to assess the paper based on its solidified technical merit.

---

### Official Review · Reviewer_RU2d · 2025-10-31

**Soundness:** 3
**Presentation:** 3
**Contribution:** 2
**Rating:** 4
**Confidence:** 4

**Summary:**

The paper introduces OBS-Diff, a training-free, one-shot pruning framework tailored for large text-to-image diffusion models, whose iterative denoising makes standard LLM-based pruning ineffective. Experiments show OBS-Diff achieves state-of-the-art one-shot pruning for diffusion models, delivering notable inference speedups with minimal drop in visual quality.

**Strengths:**

- OBS-Diff adapts OBS with a timestep-aware Hessian that gives earlier steps more weight, which matches error build-up in denoising. It also supports many sparsity types (unstructured, N:M, and structured like MHA heads and FFN neurons), making the pruning precise and flexible without extra training.
- It is a one-shot, training-free pipeline with a group-wise sequential strategy that cuts calibration cost. Experiments show clear speedups with little quality drop across common text-to-image backbones (e.g., SD/SDXL/DiT-style models), suggesting good practicality.

**Weaknesses:**

- Most comparisons are against LLM-oriented, training-free heuristics (e.g., Wanda/DSnoT/magnitude) adapted to diffusion, plus simple L1 for structured cases. This risks overstating, because these baselines were not designed around DiT denoising dynamics. The paper should add results against diffusion-native pruning on Stable Diffusion/SDXL/DiT—e.g., LD-Pruner, EcoDiff, and TinyFusion.
- The proposed timestep weighting and the group-wise pruning are tuned on a small prompt set and tested in limited settings. It’s not clear they hold for other samplers, step counts, CFG scales, or out-of-domain prompts/styles. Wider tests across these factors would better show robustness.

**Questions:**

My major concern lies in the experimental comparison. This paper only compares with LLM-based method.

---

> ### Author Response · Authors · 2025-11-23
> **Response to Reviewer RU2d (Part-1)**
>
> We sincerely thank the reviewer for the constructive feedback and for recognizing OBS-Diff as a "precise and flexible" framework that effectively matches "error build-up in denoising." We appreciate the acknowledgment of our method’s "good practicality" and "clear speedups."
>
> We have uploaded a revised PDF containing new data (highlighted in blue). Below, we address your concerns regarding comparison methods and robustness.
>
> ---
>
> >**Q1: Comparison with State-of-the-Art Diffusion Pruning (EcoDiff)**
>
> To address the concern regarding diffusion-native baselines, we carefully selected appropriate comparisons based on pruning granularity:
>
> 1. **Scope of Comparison:** OBS-Diff performs width/unstructured pruning (e.g., attention heads, FFN neurons).
>
> + TinyFusion [3] & Laptop-Diff [4] are depth pruning methods (removing entire blocks), making them structurally incomparable.
>
> + LD-Pruner [1] is closed-source, requires fine-tuning, and is limited to SD v1.4.
>
> 2. **Direct Comparison (EcoDiff):** We identified EcoDiff [2] as the most relevant state-of-the-art structured pruning baseline. We ran EcoDiff using its official implementation on SDXL (U-Net) and SD3.5-Large (MMDiT).
>
> As shown in Tables A and B, OBS-Diff consistently outperforms EcoDiff across all sparsity levels and metrics. Crucially, OBS-Diff achieves these results as a training-free, one-shot method, whereas EcoDiff requires computationally expensive mask training and hyperparameter tuning.
>
>
> **Table A: Structured Pruning on SDXL (U-Net)**
> | Model | Sparsity | Method | FID $\downarrow$ | CLIP Score $\uparrow$ | ImageReward $\uparrow$ |
> | :--- | :---: | :--- | :---: | :---: | :---: |
> | **SDXL** | **Dense** | *Original* | 29.21 | 0.3213 | 0.7635 |
> | | **15%** | L1-norm | 71.78 | 0.3035 | -0.0006 |
> | | | EcoDiff | 34.18 | 0.3100 | -0.1870 |
> | | | **OBS-Diff (Ours)** | **29.08** | **0.3215** | **0.6877** |
> | | **20%** | L1-norm | 133.07 | 0.2825 | -0.7897 |
> | | | EcoDiff | 42.98 | 0.2993 |  -0.6172 |
> | | | **OBS-Diff (Ours)** | **29.19** | **0.3212** | **0.6461** |
> | | **30%** | L1-norm | 170.68 | 0.2711 | -1.1694 |
> | | | EcoDiff | 101.96 | 0.2465 | -1.9161 |
> | | | **OBS-Diff (Ours)** | **29.75** | **0.3204** | **0.4909** |
>
>
> **Table B: Structured Pruning on SD3.5-Large (MMDiT)**
> | Model | Sparsity | Method | FID $\downarrow$ | CLIP Score $\uparrow$ | ImageReward $\uparrow$ |
> | :--- | :---: | :--- | :---: | :---: | :---: |
> | **SD 3.5-Large** | **Dense** | *Original* | 31.59 | 0.3156 | 0.7549 |
> | | **15%** | L1-norm | 158.89 | 0.2376 | -2.0502 |
> | | | EcoDiff | 230.97 | 0.2086 | -2.2594 |
> | | | **OBS-Diff (Ours)** | **32.64** | **0.3157** | **0.6446** |
> | | **20%** | L1-norm | 189.50 | 0.2124 | -2.2385 |
> | | | EcoDiff | 293.89 | 0.2050 | -2.2724 |
> | | | **OBS-Diff (Ours)** | **32.46** | **0.3149** | **0.5475** |
> | | **25%** | L1-norm | 228.82 | 0.2040 | -2.2651 |
> | | | EcoDiff | 308.96 | 0.2037 | -2.2686 |
> | | | **OBS-Diff (Ours)** | **33.73** | **0.3128** | **0.3741** |
> | | **30%** | L1-norm | 327.48 | 0.2093 | -2.2663 |
> | | | EcoDiff | 346.38 | 0.2024 | -2.2746 |
> | | | **OBS-Diff (Ours)** | **34.51** | **0.3107** | **0.2221** |

---

> ### Author Response · Authors · 2025-11-23
> **Response to Reviewer RU2d (Part-2)**
>
> > **Q2: Robustness to Inference Hyperparameters**
>
>
> To demonstrate that our calibration (on only 100 prompts) does not overfit, we evaluated the fixed pruned model (SD3-Medium, 50% Unstructured) under inference conditions significantly different from the calibration settings (CFG 7.0, Steps 25, Euler).
>
> For fast evaluation, we evaluated all tasks on MSCOCO 2014 validation 1K subset with ImageReward metric.
>
> **1. Robustness to CFG Scales.**
>
> **Table C: Robustness analysis regarding Classifier-Free Guidance (CFG) scales.**
> | CFG | Dense (ImageReward) | Pruned (ImageReward) | Performance |
> | :--- | :--- | :--- | :--- |
> | 5.0 | 0.8319 | 0.5297 | - |
> | 7.0 (Calibrated) | 0.8510 | 0.6425 | Baseline |
> | 9.0| 0.8275 | **0.7044** | **Improved** |
>
> Most notably, the pruned model achieves **higher performance at CFG 9.0 (0.7044)** than at the calibration setting of CFG 7.0 (0.6425). This indicates that our pruning strategy effectively preserves the model's semantic generation capabilities and generalizes exceptionally well to higher guidance scales, which are critical for high-quality text-to-image synthesis.
>
>
> **2. Robustness to Sampling Steps.**
>
> To address the concern regarding step counts, we evaluated the pruned model (calibrated at 25 steps) across 15, 25, and 50 inference steps. The results are summarized below:
>
> **Table D: Robustness analysis regarding inference sampling steps.**
> | Steps | Dense (ImageReward) | Pruned (ImageReward) | Trend |
> | :--- | :--- | :--- | :--- |
> | 15 | 0.6988 | 0.4883 | Fast |
> | 25 (Calibrated) | 0.8510 | 0.6425 | Baseline |
> | 50 | **0.9391** | **0.7153** | **Improved Quality** |
>
> Although calibrated at 25 steps, the pruned model effectively leverages additional compute at 50 steps to generate higher-quality images. This confirms the pruning preserves the integrity of the underlying ODE trajectory.
>
> **3. Robustness across Samplers.**
> We evaluated generalization across different solvers on both SD3-Medium (MMDiT) and SD v2.1 (U-Net).
>
> + SD3-Medium: Calibrated on Euler (1st-order), the model generalizes zero-shot to Heun (2nd-order), showing significant quality gains.
>
> + SD v2.1: We applied 40% unstructured pruning (calibrated on PNDM). As shown below, the relative performance ranking of the samplers is preserved between the Dense and Pruned models (e.g., DPM++ remains the highest performing), indicating the pruning is solver-agnostic.
>
> **Table E: Generalization of OBS-Diff across different samplers.**
> | Model | Sampler | ImageReward (Dense) | ImageReward (Pruned) |
> | :--- | :--- | :---: | :---: |
> | **SD3-Medium** | Euler (Calibrated) | 0.8510 | 0.6425 |
> | | Heun (2nd Order) | **0.9200** | **0.7249** |
> | **SD v2.1** | PNDM (Calibrated) | 0.3432 | 0.1782 |
> | | DPM++ | **0.3889** | **0.2246** |
> | | EDM | 0.3442 | 0.1534 |
> | | DDIM | 0.3439 |  0.1579 |
>
>
>
> **4. Generalization to Out-of-Distribution (OOD) Prompts**
> We address the concern regarding calibration data in two ways:
>
> 1. **Experimental Design:** All main results in the paper already use **GCC3M for calibration** and **MS-COCO for evaluation**, representing a standard OOD setting.
>
> 2. **New Validation Experiment:** To rigorously test this, we calibrated two separate models—**one using MSCOCO 2014 Train (In-Distribution)** and **one using GCC3M (Out-of-Distribution)**—and evaluated both on the **MSCOCO 2014 validation 5K subset**.
>
> **Table F: Analysis of generalization to Out-of-Distribution (OOD) calibration data.**
> | Calibration Dataset | Evaluation (on MS-COCO 5k) | FID $\downarrow$ | CLIP Score $\uparrow$ | ImageReward $\uparrow$ |
> | :--- | :--- | :--- | :--- | :--- |
> | MS-COCO 2014 Train | In-Distribution (ID) | 27.93 | 0.3169 | 0.6547 |
> | GCC3M Train | Out-of-Distribution (OOD) | 27.20 | 0.3167 | 0.6468 |
>
> The performance is nearly identical. The model calibrated on OOD data (GCC3M) performs on par with (and even slightly better in FID than) the ID model. This definitively proves that OBS-Diff captures generalizable features and does not overfit to the calibration prompts.
>
> ---

---

> ### Author Response · Authors · 2025-11-23
> **Response to Reviewer RU2d (Part-3)**
>
> We hope that the new experiments and detailed explanation successfully resolve your concerns about comparison methods and robustness. As we have strengthened the paper significantly based on your constructive feedback, we kindly ask that you reconsider your rating. Thank you again for your time and valuable insights.
>
> ---
>
> #### References
> [1] Castells, Thibault, et al. "Ld-pruner: Efficient pruning of latent diffusion models using task-agnostic insights." Proceedings of the IEEE/CVF Conference on Computer Vision and Pattern Recognition. 2024.
>
> [2] Zhang, Yang, et al. "Effortless efficiency: Low-cost pruning of diffusion models." arXiv preprint arXiv:2412.02852 (2024).
>
> [3] Fang, Gongfan, et al. "Tinyfusion: Diffusion transformers learned shallow." Proceedings of the Computer Vision and Pattern Recognition Conference. 2025.
>
> [4] Zhang, Dingkun, et al. "Laptop-diff: Layer pruning and normalized distillation for compressing diffusion models." arXiv preprint arXiv:2404.11098 (2024).

---

> ### Author Response · Authors · 2025-11-28
> **Response to Reviewer RU2d**
>
> Dear Reviewer RU2d,
>
> We hope this message finds you well. As the discussion period draws to a close, we would like to gently follow up to ensure that our new experimental results have addressed your concerns regarding baselines and robustness.
>
> 1. **Comparison with Diffusion-Native Baselines**
> + Scope Clarification: We clarified the selection of appropriate comparison methods. Methods like TinyFusion and Laptop-Diff target depth pruning (removing entire blocks), rendering them structurally incomparable to our width pruning (attention heads and FFN neurons). Additionally, LD-Pruner is closed-source, requires extensive retraining and is limited to older architectures (SD v1.4).
>
> + Direct Comparison (EcoDiff): Consequently, we focused on EcoDiff as the most relevant SOTA structured pruning baseline. We evaluated it on both SDXL and SD3.5-Large. The results (Tables A & B in our previous response) show that OBS-Diff significantly outperforms EcoDiff, especially at high sparsity levels, without requiring the training that EcoDiff needs during the pruning process.
>
> 2. **Robustness Verification** We have also conducted the requested sensitivity analyses (Tables C, D, E, F). The results confirm that our method is robust across:
>
> + CFG Scales: Generalizes well to higher guidance scales (e.g., 9.0).
>
> + Samplers: Zero-shot generalization from Euler to Heun.
>
> + OOD Calibration: Calibration on GCC3M performs on par with MS-COCO.
>
> We believe these additional experiments provide solid evidence backing the "soundness" and "practicality" of our work. Since we have directly resolved your major concern about the lack of diffusion-native comparisons, we kindly ask you to reconsider your rating.
>
> Best regards,
>
> The Authors

---

### Official Review · Reviewer_p7Tk · 2025-10-31

**Soundness:** 3
**Presentation:** 3
**Contribution:** 2
**Rating:** 6
**Confidence:** 4

**Summary:**

OBS-Diff is a training-free, one-shot pruning framework for text-to-image diffusion models. OBS-Diff employs optimized OBS for diffusion architectures and supports multiple granularities (unstructured, N:M semi-structured, and structured pruning of MHA heads / FFN neurons). A timestep-aware Hessian with decreasing weights prioritizes early denoising steps to optimize error accumulation, and a group-wise sequential procedure optimizes calibration cost. Experiments on SD3/3.5/Flux models report minimal visual quality loss with notable speedups on a wide range of sparsity types.

**Strengths:**

- Unified support for unstructured, N:M (e.g., 2:4), and structured sparsity in diffusion series models.

- Efficient group-wise sequential pruning and effective weighted Hessian construction.

- Qualitative and quantitative results across SD3-Medium/SD3.5-Large and Flux; Strong experimental metrics.

**Weaknesses:**

- Figure 2 needs to be cited and introduced at the beginning of the methods section.

- Wall-clock numbers are shown for a single MMDiT block; whole-model, sampler-inclusive speedups are not fully established. Efficiency gain under unstructured sparsity also needed to be reported.

- The comparison set is narrow, mainly Magnitude, Wanda, DSnoT for un/semi-structured pruning and an L1-norm baseline for structured pruning (with Diff-Pruning only on CIFAR-10). Please broaden or justify the baseline choice by including (or at least discussing) recent training-based / non-one-shot diffusion compression methods

- What would be the result of implementing SparseGPT like wanda? This experiment might demonstrate the advantages of the proposed approach over standard OBS.

- The log-decrease schedule is intuitive but needs stronger ablation/theory vs. alternatives (e.g., cosine, exponential, other weights).

- For CIFAR-10 DDPMs, the pruned models are fine-tuned (100k steps, C.3); clarify when fine-tuning is required vs. truly training-free regimes. Also, the enhancement of the proposed method seems to be very limited compared to other methods (even random pruning), please clarify.

- In Table 8, the higher the sparsity, the better the FID metrics, explain why.

- How sensitive is OBS-Diff to the timestep-weighting schedule and to the number/selection of timesteps used for Hessian estimation, and do the gains persist across samplers (DDIM/DPM++/EDM) and extend to video diffusion or cascaded/staged T2I? A qualitative explanation is sufficient; empirical results are optional if running new experiments would be time-consuming.

**Questions:**

See above

---

> ### Author Response · Authors · 2025-11-23
> **Response to Reviewer p7TK (Part-1)**
>
> We sincerely thank the reviewer for the thoughtful feedback and for recognizing OBS-Diff as a **"unified"** and **"efficient"** framework with **"strong experimental metrics"** across diverse architectures. We appreciate the opportunity to clarify our methodology, broaden our comparisons with state-of-the-art methods, and provide additional performance data.
>
> We have uploaded a revised PDF containing new data (highlighted in blue). Below, we address the key points raised:
>
> ---
>
> >**Q1:** Citation and placement of Figure 2.
>
> Thank you for the suggestion. We will move the citation and discussion of Figure 2 to the beginning of Section 4 (Methodology) to provide a clearer roadmap of our framework for the readers.
>
> >**Q2:** Whole-model speedup and unstructured efficiency.
>
> We report the whole-model inference latency measured on SD3.5-Large below. As shown, increasing structured sparsity leads to tangible wall-clock acceleration.
>
> We applied structured pruning to the SD3.5-Large model, excluding the first and last MMDiT blocks. All experiments were conducted on an NVIDIA RTX 4090 with a resolution of $512 \times 512$, 28 sampling steps, and a batch size of 4.
>
> **Table A: Whole-Model Inference Latency (SD3.5-Large)**
>
> | Sparsity Type | Latency (s/img) | Speedup |
> | :--- | :---: | :---: |
> | **Dense Model** | 3.97 | 1.00× |
> | **Structured (20%)** | 3.41 | 1.16× |
> | **Structured (25%)** | 3.11 | 1.28× |
> | **Structured (30%)** | 3.10 | 1.28× |
>
> **Regarding Unstructured Pruning:** While unstructured pruning does not directly translate to hardware acceleration without specialized kernels, it serves a critical scientific purpose. It acts as the "gold standard" for measuring **inherent model redundancy**, establishing the theoretical upper bound for compressibility. Furthermore, the algorithmic success of OBS-Diff in the unstructured regime provides the mathematical foundation that enables our high-performance extension to hardware-friendly granularities, such as N:M semi-structured and structured pruning (removing entire MHA heads and FFN neurons).
>
> >**Q3:** Broadening baseline comparisons.
>
> 1. **Scope of Comparison:** OBS-Diff performs width/unstructured pruning (e.g., attention heads, FFN neurons).
>
> + TinyFusion [3] & Laptop-Diff [4] are depth pruning methods (removing entire blocks), making them structurally incomparable.
>
> + LD-Pruner [1] is closed-source, requires fine-tuning, and is limited to SD v1.4.
>
> 2. **Direct Comparison (EcoDiff):** We identified EcoDiff [2] as the most relevant state-of-the-art structured pruning baseline. We ran EcoDiff using its official implementation on SDXL (U-Net) and SD3.5-Large (MMDiT).
>
> As shown in Tables A and B, OBS-Diff consistently outperforms EcoDiff across all sparsity levels and metrics. Crucially, OBS-Diff achieves these results as a training-free, one-shot method, whereas EcoDiff requires computationally expensive mask training and hyperparameter tuning.
>
> We follow the official manuscipt of EcoDiff at github, applied it on both U-Net-based model (SDXL) and MMDiT-based model (SD3.5-large), and here is the result.
>
> **Table B: Structured Pruning on SDXL (U-Net)**
> | Model | Sparsity | Method | FID $\downarrow$ | CLIP Score $\uparrow$ | ImageReward $\uparrow$ |
> | :--- | :---: | :--- | :---: | :---: | :---: |
> | **SDXL** | **Dense** | *Original* | 29.21 | 0.3213 | 0.7635 |
> | | **15%** | L1-norm | 71.78 | 0.3035 | -0.0006 |
> | | | EcoDiff | 34.18 | 0.3100 | -0.1870 |
> | | | **OBS-Diff (Ours)** | **29.08** | **0.3215** | **0.6877** |
> | | **20%** | L1-norm | 133.07 | 0.2825 | -0.7897 |
> | | | EcoDiff | 42.98 | 0.2993 |  -0.6172 |
> | | | **OBS-Diff (Ours)** | **29.19** | **0.3212** | **0.6461** |
> | | **30%** | L1-norm | 170.68 | 0.2711 | -1.1694 |
> | | | EcoDiff | 101.96 | 0.2465 | -1.9161 |
> | | | **OBS-Diff (Ours)** | **29.75** | **0.3204** | **0.4909** |
>
> **Table C: Structured Pruning on SD3.5-Large (MMDiT)**
> | Model | Sparsity | Method | FID $\downarrow$ | CLIP Score $\uparrow$ | ImageReward $\uparrow$ |
> | :--- | :---: | :--- | :---: | :---: | :---: |
> | **SD 3.5-Large** | **Dense** | *Original* | 31.59 | 0.3156 | 0.7549 |
> | | **15%** | L1-norm | 158.89 | 0.2376 | -2.0502 |
> | | | EcoDiff | 230.97 | 0.2086 | -2.2594 |
> | | | **OBS-Diff (Ours)** | **32.64** | **0.3157** | **0.6446** |
> | | **20%** | L1-norm | 189.50 | 0.2124 | -2.2385 |
> | | | EcoDiff | 293.89 | 0.2050 | -2.2724 |
> | | | **OBS-Diff (Ours)** | **32.46** | **0.3149** | **0.5475** |
> | | **25%** | L1-norm | 228.82 | 0.2040 | -2.2651 |
> | | | EcoDiff | 308.96 | 0.2037 | -2.2686 |
> | | | **OBS-Diff (Ours)** | **33.73** | **0.3128** | **0.3741** |
> | | **30%** | L1-norm | 327.48 | 0.2093 | -2.2663 |
> | | | EcoDiff | 346.38 | 0.2024 | -2.2746 |
> | | | **OBS-Diff (Ours)** | **34.51** | **0.3107** | **0.2221** |
>
> From these two tables, we can observe that OBS-Diff outperform the EcoDiff on the all aspects.

---

> ### Author Response · Authors · 2025-11-23
> **Response to Reviewer p7TK (Part-2)**
>
> >**Q4:** Comparison with SparseGPT and advantages.
>
> We appreciate this insightful question. To address this, we must first clarify the relationship between SparseGPT and our proposed OBS-Diff. Both methods are rooted in the classic Optimal Brain Surgeon (OBS) theory, which utilizes Hessian information to prune redundant weights and compensate for the remaining ones.
>
> However, directly applying LLM-centric methods like SparseGPT or Wanda to diffusion models is non-trivial due to the **iterative denoising process**. This introduces two challenges: (1) the prohibitive cost of standard calibration across multiple timesteps, and (2) the error accumulation across the iterative denoising steps.
>
> To bridge this gap, we introduced the "Module Package" strategy to make calibration feasible. Consequently, implementing "SparseGPT for Diffusion" is mathematically equivalent to our OBS-Diff framework using a "Uniform" timestep weighting scheme (i.e., treating all timesteps as equally important during Hessian estimation).
>
> We have actually evaluated this exact scenario in our ablation study (**Table 6** of the main paper). The comparison below demonstrates that while the adapted SparseGPT is effective, our proposed Timestep-Aware Hessian (Log-decrease) yields superior performance by specifically addressing the error accumulation problem inherent in diffusion models.
>
> **Table D: Comparison with Adapted SparseGPT.**
> | Method | ImageReward (MSCOCO 2014 validation 1K) |
> | :--- | :--- |
> | adapted-SparseGPT (module-package) | 0.6355 |
> | OBS-Diff (module-package + timestep-aware Hessian) | 0.6438 |
> >**Q5:** Justification for the log-decrease schedule.
>
> While exploring additional schedules (e.g., cosine, exponential) is theoretically interesting, it introduces significant experimental overhead and hyperparameter tuning costs. We believe our current ablation in Table 6 is sufficient to establish the core design principle:
>
> + **Directionality:** Decreasing schedules consistently outperform increasing ones, confirming that assigning higher weights to early timesteps is essential to mitigate error accumulation.
>
> + **Profile:** The Log-decrease schedule outperforms the Linear-decrease strategy, validating that a rapid initial emphasis followed by a "long tail" aligns better with the denoising dynamics than linear weighting.
>
> Since we have already identified a effective strategy that validates our Timestep-Aware framework, we consider the search for a globally optimal weighting function to be a valuable follow-up optimization problem, rather than a prerequisite for the current contributions.
>
> >**Q6:** Clarification on Fine-tuning and DDPM gains.
>
> 1. **Scope:** Our primary contribution—compression of large-scale text-to-image models—is entirely training-free. The CIFAR-10 experiment serves strictly to benchmark against Diff-Pruning, a method that relies on fine-tuning, and is applied on class-conditional diffusion model rather than our t2i diffusion model tasks. We applied fine-tuning solely to ensure a fair, apples-to-apples comparison.
>
> 2. **Enhancement:** Given that CIFAR-10 DDPM is a small, saturated model (35.7M parameters), the gains are significant. OBS-Diff (FID 7.55) outperforms both Random Pruning (7.85) and the previous SOTA Diff-Pruning (7.72) under the exact same budget, proving our Hessian criterion provides a superior initialization.

---

> ### Author Response · Authors · 2025-11-23
> **Response to Reviewer p7TK (Part-3)**
>
> >**Q7:** Inverse trend of FID in Table 8.
>
> 1. **Data Correction for Dense Model**: Before addressing the trend, we must first rectify a clerical error identified in Table 8 regarding the "Dense Model" baseline.
> We inadvertently copied the "Dense Model" data from the SD3.5-Large experiment (Table 1b/Table 3) instead of the correct SD3-Medium “Dense Model" (from Table 1a). We have updated the table below with the correct Dense metrics (FID: 36.14 CLIP 0.3162 ImageReward 0.3627). We sincerely apologize for this oversight.
>
> **Table E: Corrected metrics for SD3-Medium Structured Pruning.**
> | Base Model | Sparsity (%) | Method | FID $\downarrow$ | CLIP $\uparrow$ | ImageReward $\uparrow$ |
> | :--- | :--- | :--- | :--- | :--- | :--- |
> | | | **Dense Model** | **36.14** | **0.3162** | **0.9029** |
> | **SD 3-medium** | 10% | $L_1 - norm$ | 267.32 | 0.2035 | -2.2611 |
> | | | Ours (OBS-diff) | **35.65** | **0.3166** | **0.8118** |
> | | 15% | $L_1 - norm$ | 326.92 | 0.1942 | -2.2768 |
> | | | Ours (OBS-diff) | **34.33** | **0.3168** | **0.6717** |
> | | 20% | $L_1 - norm$ | 348.77 | 0.1926 | -2.2768 |
> | | | Ours (OBS-diff) | **33.15** | **0.3163** | **0.4997** |
> | | 25% | $L_1 - norm$ | 365.24 | 0.1906 | -2.2786 |
> | | | Ours (OBS-diff) | **32.96** | **0.3143** | **0.2782** |
>
> 2. **Explanation of the Inverse FID Trend:**
>
> From the corrected Table 8, we observe that the pruned models achieve lower (better) FID scores than the Dense model, and FID continues to decrease as sparsity increases. This counter-intuitive phenomenon aligns with observations in Table 1 and can be attributed to the inherent limitations of the FID metric itself.
>
> + **Misalignment with perceptual quality:** It is well-documented that FID often **decouples from human perceptual judgments.** As argued in Rethinking FID (CVPR 2024) [6], FID *"contradicts human raters, does not reflect gradual improvement of iterative text-to-image models, it does not reflect gradual improvement of iterative text-toimage models, it does not capture distortion levels, and that it produces inconsistent results when varying the sample size. "* In the context of pruning, the removal of certain weights may inadvertently suppress high-frequency artifacts that the Inception-v3 network (used for FID) penalizes, artificially boosting the score despite a loss in semantic detail.
>
> + **Evidence from Reliable Metrics:** Crucially, we urge the reviewer to prioritize ImageReward [5], which are known to align better with human aesthetic preferences and semantic consistency.
> As shown in the corrected table, ImageReward exhibits a strict monotonic decrease (0.9029 $\to$ 0.8118 $\to$ ... $\to$ 0.2782) as sparsity increases. This correctly reflects the expected trade-off between compression rate and model capacity. This confirms that the "improvement" in FID is an artifact of the metric, not the model.
>
> We once again apologize for this oversight. Following this, we have conducted a comprehensive review of all numerical results presented in the manuscript against our raw experimental logs to ensure accuracy. We confirm that this was an isolated clerical error, and all other reported data remain correct.

---

> ### Author Response · Authors · 2025-11-23
> **Response to Reviewer p7TK (Part-4)**
>
> >**Q8:** Sensitivity to samplers and extension to video.
>
>
> **1. Sensitivity to Timestep Settings**
> * **Weighting Schedule:** As detailed in **Table 6**, OBS-Diff exhibits sensitivity to the temporal weighting profile. Specifically, "decreasing" schedules (prioritizing early steps) consistently outperform uniform or increasing strategies, with "Log-decrease" achieving the optimal performance. This empirically validates our hypothesis that minimizing error accumulation in early denoising steps is critical.
> * **Number of Estimation Timesteps:** We further evaluated the impact of the number of timesteps used to accumulate Hessian statistics. As shown below, performance improves consistently as temporal resolution increases. While our default setting (25 steps) strikes an optimal balance between calibration cost and quality, increasing to 50 steps yields further gains (ImageReward 0.6581), demonstrating that the method is stable and scalable.
>
> **Table F: Sensitivity to the number of Hessian estimation timesteps.**
> | Estimation Steps | 15 | **25 (Default)** | 50 |
> | :--- | :---: | :---: | :---: |
> | **ImageReward** | 0.6308 | **0.6425** | 0.6581 |
>
> **2. Robustness across Samplers.**
> We evaluated generalization across different solvers on both SD3-Medium (MMDiT) and SD v2.1 (U-Net).
>
> + SD3-Medium: **Calibrated on Euler** (1st-order), the model generalizes zero-shot to Heun (2nd-order), showing significant quality gains.
>
> + SD v2.1: We applied 40% unstructured pruning (**calibrated on PNDM**). As shown below, the relative performance ranking of the samplers is preserved between the Dense and Pruned models (e.g., DPM++ remains the highest performing), indicating the pruning is solver-agnostic.
>
>
> **Table G: Generalization across Samplers.**
> | Model | Sampler | ImageReward (Dense) | ImageReward (Pruned) |
> | :--- | :--- | :---: | :---: |
> | **SD3-Medium** | Euler (Calibrated) | 0.8510 | 0.6425 |
> | | Heun (2nd Order) | **0.9200** | **0.7249** |
> | **SD v2.1** | PNDM (Calibrated) | 0.3432 | 0.1782 |
> | | DPM++ | **0.3889** | **0.2246** |
> | | EDM | 0.3442 | 0.1534 |
> | | DDIM | 0.3439 |  0.1579 |
>
> **3. Extension to Video and Cascaded Models:** We truly appreciate the reviewer's consideration regarding the time constraints of running additional experiments.
>
> Theoretically, OBS-Diff is architecture-agnostic. The main contributions of our work—**the Module-Package strategy** (which renders iterative calibration computationally feasible) and **the Timestep-Aware Hessian** (which addresses error accumulation)—address the fundamental challenges of pruning **diffusion models**. Consequently, OBS-Diff can be readily extended to video diffusion or cascaded models by simply adapting the calibration dataset (e.g., using video prompts). The framework's ability to identify structural redundancy remains effective regardless of the specific modality.
>
> ---
>
> We sincerely appreciate your detailed review. We hope our responses have clarified your main concerns. Thank you again for helping us improve this work.
>
>
>
>
> References
>
> [1] Castells, Thibault, et al. "Ld-pruner: Efficient pruning of latent diffusion models using task-agnostic insights." Proceedings of the IEEE/CVF Conference on Computer Vision and Pattern Recognition. 2024.
>
> [2] Zhang, Yang, et al. "Effortless efficiency: Low-cost pruning of diffusion models." arXiv preprint arXiv:2412.02852 (2024).
>
> [3] Fang, Gongfan, et al. "Tinyfusion: Diffusion transformers learned shallow." Proceedings of the Computer Vision and Pattern Recognition Conference. 2025.
>
> [4] Zhang, Dingkun, et al. "Laptop-diff: Layer pruning and normalized distillation for compressing diffusion models." arXiv preprint arXiv:2404.11098 (2024).
>
> [5] Xu, Jiazheng, et al. "Imagereward: Learning and evaluating human preferences for text-to-image generation." Advances in Neural Information Processing Systems 36 (2023): 15903-15935.
>
> [6] Jayasumana, Sadeep, et al. "Rethinking fid: Towards a better evaluation metric for image generation." Proceedings of the IEEE/CVF Conference on Computer Vision and Pattern Recognition. 2024.

---

### Official Review · Reviewer_L2zh · 2025-11-01

**Soundness:** 3
**Presentation:** 3
**Contribution:** 2
**Rating:** 2
**Confidence:** 4

**Summary:**

This paper introduces OBS-Diff, a one-shot, training-free pruning framework designed to compress large-scale text-to-image diffusion models. The authors identify that existing methods fail because they don't account for the iterative nature of the denoising process, where errors can accumulate. OBS-Diff addresses this by adapting the classic Optimal Brain Surgeon (OBS) method. Its core innovations are a timestep-aware Hessian construction that gives more weight to earlier, more critical denoising steps, and a group-wise pruning strategy to make the process computationally efficient.

**Strengths:**

1. The proposed timestep-aware Hessian is an elegant and intuitive solution. Weighting the pruning criteria based on the temporal dynamics of the diffusion process is a clever way to mitigate performance degradation.

2. The framework is explicitly designed to handle modern, complex architectures (like MMDiT) and supports a wide range of pruning granularities (unstructured, semi-structured, and structured). This makes it a potentially universal tool for diffusion model compression.

**Weaknesses:**

1. the main result (Table 1) presents the unstructured pruning results. However, unstructured pruning have limited application and cannot provide memory reduction and speedup on conventional hardware. From the table, models can be pruned up to 50%. Similar results have been presented by baseline works (SparseGPT, Wanda), which shows the redundancy of transformer weights. Therefore, showing diffusion models can be pruned by 50% in an unstructured way has limited contribution.
2. Comparison results are mainly for unstructured pruning. Authors should show comparison results in structured pruning and semi-structured pruning settings to support the advantage of OBS-Diff in a practical scenario. From Table 2, Wanda is better than OBS-Diff under semi-structured setting. Comparison under structured setting is missing.
3. While many models are used under unstructured setting. Only SD3.5 is used under semi-structured and structured setting.

**Questions:**

Same as above.

---

> ### Author Response · Authors · 2025-11-23
> **Response to Reviewer L2zh (Part-1)**
>
> We thank the reviewer for recognizing our Timestep-Aware Hessian as "elegant" and "intuitive", and for acknowledging OBS-Diff as a potential "universal tool" for modern architectures.
>
> We respectfully disagree with the assessment that our contribution is limited. Below, we clarify the unique challenges of diffusion pruning and provide new semi-structured and structured comparisons (including **both UNet-based and MMDiT-based** models) that demonstrate OBS-Diff's clear superiority over baselines like Wanda and EcoDiff.
>
> ---
>
> >**Q1**: The Scientific Value of Unstructured Pruning in Diffusion
>
> We respectfully disagree with the assertion that our unstructured pruning on diffusion models have limited contributions. We present three key arguments regarding its scientific value
>
> **1. Unstructured pruning serves as the theoretical upper bound.** While structured pruning is hardware-friendly, unstructured pruning is widely recognized in the research community as the "gold standard" for **measuring algorithm effectiveness and inherent model redundancy**. It provides the theoretical upper bound of compressibility.
>
> **2. Diffusion Redundancy $\neq$ LLM Redundancy.** The reviewer suggests that because Transformer-based LLMs are redundant (as shown by Wanda/SparseGPT), diffusion models are trivially so. This is a misconception.
> Although both share the Transformer architecture, the established redundancy of **autoregressive** LLMs does not imply that **iterative** diffusion models are easily pruned.
> OBS-Diff employs **Module Packages** and **Timestep-Aware Hessian construction** to specifically address *the unique challenges posed by the iterative diffusion process.* By successfully pruning text-to-image diffusion models by 50% in an unstructured manner, we empirically demonstrate that diffusion weights indeed possess significant redundancy, a finding enabled by our targeted methodology rather than a trivial property.
>
> **3. Extend unstructured pruning to semi and structured guanlarity.** With the well-established algorithm of unstructured pruning guanlarity, we easily extended to further hardware-friendly pruning guanlarity, such as semi-structured pruning and structured pruning, and obtained the state-of-the-art result.
>
> ---
>
> >**Q2:** Performance on Semi-Structured Pruning (Addressing "Wanda is better than OBS-Diff)
>
> The reviewer noted that Wanda appeared better in Table 2. We clarify that the FID difference (0.05) was negligible, while OBS-Diff significantly led in semantic and human preference metrics (CLIP/ImageReward).
>
> To strictly demonstrate superiority, we extended the pruning range (blocks 3-26 and 3-27) on SD3.5-Large. As shown below, OBS-Diff consistently outperforms Wanda on all metrics, including FID.
>
> **Table A: Further Semi-Structured Pruning on SD3.5-Large**
> | Base Model | Pruning Block Range | Method | FID ↓ | CLIP ↑ | ImageReward ↑ |
> | :--- | :--- | :--- | :---: | :---: | :---: |
> | **SD 3.5-Large** | 3-25 | Wanda | **32.08** | 0.3036 | -0.1363 |
> | | | OBS-Diff | 32.13 | **0.3129** | **0.4493** |
> | | 3-26 | Wanda | 32.33 | 0.3019 | -0.1509 |
> | | | OBS-Diff | **32.05** | **0.3127** | **0.3818** |
> | | 3-27 | Wanda | 34.79 | 0.2979 | -0.2958 |
> | | | OBS-Diff | **31.20** | **0.3124** | **0.3778** |
>
> We also apply OBS-Diff on the U-Net based text-to-image model SDv2.1, to show the generalization ability on different architecture.
>
>
> **Table B: Semi-Structured Pruning on SDv2.1 (U-Net)**
> | Model | Method | FID ↓ | CLIP ↑ | ImageReward ↑ |
> | :--- | :--- | :--- | :--- | :--- |
> | **SD v2.1-base** | Magnitude | 150.58 | 0.2329 | -2.0492 |
> | | DSnoT | 145.53 | 0.2354 | -2.0562 |
> | | Wanda | 115.08 | 0.2561 | -1.6918 |
> | | OBS-Diff | **33.33** | **0.3056** | **-0.2710** |
>
> OBS-Diff significantly outperforms other methods.

---

> ### Author Response · Authors · 2025-11-23
> **Response to Reviewer L2zh (Part-2)**
>
> >**Q3:** Performance on Structured Pruning
>
> Due to the time limit and experiment cost, we do the structured pruning on two representative model -- SDXL (**U-Net**) and SD3.5-large (**MMDiT**), comparing our OBS-Diff method performance with L1-norm pruning and the state-of-the-art structured method EcoDiff。
>
> **Table C: Structured Pruning on SDXL (U-Net)**
> | Model | Sparsity | Method | FID $\downarrow$ | CLIP Score $\uparrow$ | ImageReward $\uparrow$ |
> | :--- | :---: | :--- | :---: | :---: | :---: |
> | **SDXL** | **Dense** | *Original* | 29.21 | 0.3213 | 0.7635 |
> | | **15%** | L1-norm | 71.78 | 0.3035 | -0.0006 |
> | | | EcoDiff | 34.18 | 0.3100 | -0.1870 |
> | | | **OBS-Diff (Ours)** | **29.08** | **0.3215** | **0.6877** |
> | | **20%** | L1-norm | 133.07 | 0.2825 | -0.7897 |
> | | | EcoDiff | 42.98 | 0.2993 |  -0.6172 |
> | | | **OBS-Diff (Ours)** | **29.19** | **0.3212** | **0.6461** |
> | | **25%** | L1-norm | 189.23 | 0.2592 |-1.4452 |
> | | | EcoDiff | 87.23 | 0.2658 | -1.5778 |
> | | | **OBS-Diff (Ours)** | **29.45** | **0.3208** | **0.5857** |
> | | **30%** | L1-norm | 170.68 | 0.2711 | -1.1694 |
> | | | EcoDiff | 101.96 | 0.2465 | -1.9161 |
> | | | **OBS-Diff (Ours)** | **29.75** | **0.3204** | **0.4909** |
>
> **Table D: Structured Pruning on SD3.5-Large (MMDiT)**
> | Model | Sparsity | Method | FID $\downarrow$ | CLIP Score $\uparrow$ | ImageReward $\uparrow$ |
> | :--- | :---: | :--- | :---: | :---: | :---: |
> | **SD 3.5-Large** | **Dense** | *Original* | 31.59 | 0.3156 | 0.7549 |
> | | **15%** | L1-norm | 158.89 | 0.2376 | -2.0502 |
> | | | EcoDiff | 230.97 | 0.2086 | -2.2594 |
> | | | **OBS-Diff (Ours)** | **32.64** | **0.3157** | **0.6446** |
> | | **20%** | L1-norm | 189.50 | 0.2124 | -2.2385 |
> | | | EcoDiff | 293.89 | 0.2050 | -2.2724 |
> | | | **OBS-Diff (Ours)** | **32.46** | **0.3149** | **0.5475** |
> | | **25%** | L1-norm | 228.82 | 0.2040 | -2.2651 |
> | | | EcoDiff | 308.96 | 0.2037 | -2.2686 |
> | | | **OBS-Diff (Ours)** | **33.73** | **0.3128** | **0.3741** |
> | | **30%** | L1-norm | 327.48 | 0.2093 | -2.2663 |
> | | | EcoDiff | 346.38 | 0.2024 | -2.2746 |
> | | | **OBS-Diff (Ours)** | **34.51** | **0.3107** | **0.2221** |
>
> The result shows that OBS-Diff outperforms the comparision method L1-norm and EcoDiff, maintaining high visual performance after structured pruning.
>
> ---
>
> ## Conclusion
>
> We have addressed the reviewer's primary concerns regarding **the contribution of unstructured pruning** and the **practical advantages** of OBS-Diff.
>
> 1. **Scientific Value**: The established redundancy in LLMs does not imply the same for diffusion models. OBS-Diff successfully demonstrates that diffusion weights possess significant redundancy (up to 50% unstructured sparsity), a result achieved specifically through our novel Module Packages and Timestep-Aware Hessian construction.
>
> 2. **Practicality**: We conducted extensive comparisons for **semi-structured** and **structured pruning** on **both U-Net and DiT architectures**, where OBS-Diff consistently outperforms baselines. Concrete speedups and memory reductions are reported in the tables, with intuitive visual comparisons provided in the appendix.
>
> ---
> We have uploaded a revised PDF containing new data (highlighted in blue).
>
> We believe our detailed response and supplementary experiments highlight the unique value of our method. Given OBS-Diff's demonstrated effectiveness as a unified framework across unstructured, semi-structured, and structured settings, we respectfully request that you reconsider your evaluation and raise the score.

---

> > ### Comment · Reviewer_L2zh · 2025-11-25
> >
> > Thanks for your response. It is very surprising that one-shot structural pruning without finetuning can achieve up to 30% sparsity on models such as SD-XL and SD3.5, as all other structural pruning baselines (LD-Pruner, EcoDiff, BK-SDM) require considerable amount of post-prune finetune to achieve comparable or even lower sparsity.
> >
> > Before reconsidering my score, I would like further clarification regarding the pruned model. Can you provide pruned model weights in an anonymous link?

---

> > > ### Author Response · Authors · 2025-11-27
> > > **Response to Reviewer L2zh**
> > >
> > > We thank the reviewer for the feedback. To address your concern and facilitate direct verification, we have provided the pruned model weights via the anonymous link below:
> > >
> > > 1.https://modelscope.cn/models/OBSdiff/OBS-Diff-SDXL
> > >
> > > 2.https://modelscope.cn/models/OBSdiff/OBS-Diff-SD3.5-Large
> > >
> > > These two repositories contain the following checkpoints, all achieved without fine-tuning:
> > > 1. SD-XL: 10%, 15%, 20%, 25%, and 30% sparsity.
> > > 2. SD3.5-Large: 15%, 20%, 25%, and 30% sparsity.
> > >
> > > We believe these artifacts solidly substantiate our experimental claims. Since we have provided the requested evidence to resolve your primary concern, we kindly ask you to reconsider your score.

---

> ### Author Response · Authors · 2025-11-28
> **Response to Reviewer L2zh**
>
> Dear Reviewer L2zh,
>
> Thank you for your continued engagement. We appreciate you highlighting that our results are "surprising". We completely understand this reaction, given that successful structural pruning without fine-tuning has been a significant challenge in previous works.
>
> To substantiate our experimental findings and address your concern, **we have provided the full suite of structured pruning weights (covering multiple sparsity levels for both SD-XL and SD3.5-Large)** in our previous comment. These checkpoints serve as direct evidence that the reported performance is robust and reproducible.
>
> Since we have fulfilled your request by providing the artifacts that verify these results, we believe the primary concern regarding the validity of our method has been resolved. We kindly ask you to reconsider your assessment.
>
> Best regards,
>
> The Authors

---

### Author Response · Authors · 2025-11-30
**Summary of Reviews and Rebuttal Process**

**To ACs:**

Thanks for your service to ICLR26! We respectfully submit this brief summary to assist you in evaluating our paper.

## 1. Summary of our paper

**Goal:** We bridge the gap between the classic OBS pruning framework and Diffusion Models, addressing the challenges posed by the iterative denoising process.

**Methodology:**
+ Timestep-aware Hessian to prioritize early steps and mitigate error accumulation.

+ Efficient module-package to significantly reduce calibration costs.

OBS-Diff serves as a **unified, training-free** framework supporting multiple pruning granularities across diverse architectures.

**Results:**

1. We first systematically evaluate the unstructured and semi-structured pruning algorithm in text-to-image diffusion models (both UNet and MMDiT architectures).
2. Our structured pruning **outperforms the SOTA width pruning framework EcoDiff**, both on UNet (SDXL) and MMDiT (SD3.5-Large) architectures, while remaining **completely training-free**, whereas EcoDiff requires computationally expensive mask training.

## 2. Reviews Overview

At the initial state, we received scores of **6 / 6 / 4 / 2**. Until now, the only reviewer who responded to our rebuttal is Reviewer `L2zh` (the most negative one, with rating 2). The other three reviewers have **not** responded since we posted the rebuttals on November 23.

## 3. Response to Reviewer L2zh (rating: 2)

Reviewer L2zh's main concerns are 2 points: **1) the value of unstructured pruning, 2)** **requesting comparisons on semi-structured and structured pruning to show the practical acceleration of our method.**

### **Our Actions:**

- **On Scientific Value:** It is a consensus in the area that unstructured pruning achieves higher performance than semi, structured at the same total sparsity. While acknowledging hardware constraints, unstructured pruning serves as a standard **'touchstone'** for **evaluating algorithmic validity.** We report these results to ensure fair methodological comparison, rather than to demonstrate immediate hardware efficiency.
- **As for practical values:** We provided new experiments on semi-structured and structured pruning, comparing our method with SOTA methods like **Wanda** and **EcoDiff** across UNet (SDv2.1 or SDXL) and MMDiT (SD3.5-Large) architectures. The results show that OBS-Diff outperforms all baselines across tasks, especially at higher sparsity levels. We also provided the actual inference acceleration in Table 5 and visual performance in appendix.

### **Reviewer L2zh's Feedback:**

Upon reviewing our comparison results, Reviewer `L2zh` replied: *"It is very surprising that one-shot structural pruning without finetuning can achieve up to 30% sparsity on models such as SD-XL and SD3.5, as all other structural pruning baselines (LD-Pruner, EcoDiff, BK-SDM) require considerable amount of post-prune finetune to achieve comparable or even lower sparsity.."*

### **Our Follow-up:**

To address this concern, we provided the **pruned model weights** *(All checkpoints achieved without fine-tuning)* to substantiate our experimental findings.

**Reproducibility Links:**

- **SD-XL:** https://modelscope.cn/models/OBSdiff/OBS-Diff-SDXL
- **SD3.5-Large:** https://modelscope.cn/models/OBSdiff/OBS-Diff-SD3.5-Large

**Summary of the rebuttal to reviewer L2zh:**

The reviewer agrees that our results are strong (”very surprising”) and our uploaded checkpoints should fully address his/her concern.

---

## 4. Reviewer RU2d (rating: 4)

The concerns the reviewer `RU2d` largely focused on two areas:

1. **Comparison with SOTA diffusion pruning methods.**
2. **Robustness** across factors such as samplers, OOD prompts, dataset size, steps, and CFG.

**Our Explanations:**

- **Comparison:** We detailed the comparison with the SOTA width pruning framework **EcoDiff**. We tested on SDXL (UNet) and SD3.5-Large (MMDiT), showing OBS-Diff outperforms it, especially at higher sparsity. Crucially, OBS-Diff is **totally training-free**, whereas EcoDiff requires training during the pruning process.
- **Robustness:** We conducted comprehensive new experiments on samplers, OOD prompts, dataset size, steps, and CFG to demonstrate the robustness of our method.

---

## 5. Reviewer p7Tk and ePsz.

Besides the same concerns mentioned above, they have some minor concerns, such as the comparison of inference-time acceleration and the possible extension to other diffusion-based models. We also give detailed explanations or new experiments to clarify.

---

## 6. Conclusion

We thank all the reviewers and ACs for improving our paper. The comments have been well taken and reflected in our revised paper (see the highlighted content in blue in the PDF).

We believe that the extensive new experiments, detailed clarifications, and the released model weights have effectively addressed the concerns raised by the reviewers.

Thanks for your precious time, ACs!

Best,

Authors

---

### Meta-Review · Area_Chair_9Juk · 2026-01-05

**Summary:**

Reviewer L2zh raised concerns on the limited practical applicability of unstructured pruning focused in this paper and suggested evaluations on semi-structured and structured pruning. Authors have have provided reasonable explanations and conducted additional experiments on structured pruning, and shown improvements over other methods such as EcoDiff. L2zh further asked the pruned model weights, and authors have provided them too.

Reviewer RU2d mainly raised concerns on the lack of experiments with diffusion model pruning methods. Authors have responded by sharing comparisons to EcoDiff (one of the methods suggested by the reviewer) with convincing results.

Reviewer p7TK and ePsz are overall positive and asked multiple clarification questions and authors have provided detailed rebuttal.

**Reviewer Concerns:**

I think most of reviewers' concerns are addressed by the rebuttal.

**Reviewer Scores:**

L2zh and RU2d might have increased the score a bit if he had seen the last responses.
p7TK and ePsz may have kept their scores of 6.

---

### Decision · Program_Chairs · 2026-01-26

Accept (Poster)